# Numerical Modeling of Ice Detachment Tipping Processes: Insights from the Sedongpu Glacier, Southeastern Tibetan Plateau

Tong Zhang[1,2], Wei Yang[2], Yuzhe Wang[3], Chuanxi Zhao[4], Qingyun Long[1], and Cunde Xiao[1]

[1]State Key Laboratory of Earth Surface Processes and Disaster Risk Reduction, Faculty of Geographical Science, Beijing Normal University, Beijing 100875, China
[2]State Key Laboratory of Tibetan Plateau Earth System, Environment and Resources (TPESER), Institute of Tibetan Plateau Research, Chinese Academy of Sciences, Beijing, 100101, China
[3]College of Geography and Environment, Shandong Normal University, Jinan, China, 250014
[4]School of Water Conservancy and Environment, University of Jinan, Jinan, China, 250001

**Correspondence:** Tong Zhang (tzhang@bnu.edu.cn) and Wei Yang (yangww@itpcas.ac.cn)

**Abstract.** Glacier detachment is a severe natural hazard that can cause enormous damage in downstream regions. During detachment, a glacier will experience an abrupt change from slow-moving to high-speed flow within minutes. In this study, we investigate a massive glacier detachment event that occurred in 2018 in the Sedongpu Valley, southeastern Tibet, using a two-dimensional first-order ice flow model incorporating a positive feedback mechanism between ice stiffness and basal slip. In this model, detachment can be triggered if the ice stress exceeds the initial yield strength of glacier ice. By including this tipping mechanism, we simulate abrupt changes in the ice flow pattern of the Sedongpu Glacier. The transition from slow to abrupt flow occurs after most regions of the glacier reach a plastic state. The modeled duration of the 2018 Sedongpu detachment is comparable with observations. The abrupt weakening of ice strength during the transition from elastic to plastic deformation may be a primary cause of glacier detachment tipping processes.

## 1 Introduction

Glacier avalanche/detachment is one of the most catastrophic natural disasters in mountainous regions and serves as clear evidence of tipping elements in the cryosphere. Recent global warming trends have intensified, increasing glacier instability and the probability of ice avalanche/detachment events, thereby posing significant risks to downstream populations and infrastructure (Acharya et al., 2023; Zhang et al., 2024). For example, in the 1950s, Zelongnong and Guxianggou ice avalanches caused river-blocking disasters in Tibet (Hu et al., 2018). In 1962, Peru's Huascarán ice avalanche and subsequent debris flow devastated the Andes Mountains (Salzmann et al., 2004). In 2002, Russia's Kolka ice avalanche triggered a mudslide (Kotlyakov et al., 2004). Between 2009 and 2016, ice avalanche-rockfall events on the Siachen Glacier, Himalaya, resulted in fatalities (Berthier and Brun, 2019). In 2016, a massive twin-glacier collapse occurred in Aru, Tibet (Gilbert et al., 2018; Kääb et al., 2018). Most notably, in October 2018, the Sedongpu Glacier detached, releasing nearly 130 million m³ of ice-debris mass and blocking the Yarlung Tsangpo River for two days, which threatened downstream regions (including Bangladesh) with flooding (Li et al., 2022).

However, the dynamic mechanism underlying ice avalanche/detachment remains unclear. Previously, Gilbert et al. (2020) and Kääb et al. (2018) concluded that–among a combination of climatological, glaciological, and geomorphological triggers–deformable beds and changes in basal friction were key factors responsible for the Aru ice avalanches. Subsequently, Bai and He (2020) used seismic wave observations to estimate glacier motion parameters and simulate the Aru avalanche's extent. Nevertheless, a clear and in-depth physical and numerical explanation for the abrupt transient behavior of glacier detachment is still lacking. This gap presents a significant challenge in developing effective early warning systems for damage control and risk management.

Previously, a well-studied fracture criterion that defines relationships between material strength and applied stresses has been widely applied in glaciology to model ice fracture and iceberg calving, as well as in studies of ice flow mechanics (Pralong and Funk, 2005; Albrecht and Levermann, 2012; Duddu and Waisman, 2012). Most numerical ice flow models adopt a stress threshold approach, where fracture occurs when stresses exceed a critical value (Hulbe et al., 2010; Borstad et al., 2016; Jiménez et al., 2017), though alternative methods like pressure or strain thresholds (Duddu et al., 2020) remain less utilized. Despite laboratory benchmarks, natural system observations to validate fracture criteria and stress thresholds remain scarce.

Glacier fracture and damage significantly accelerate ice flow by structurally weakening ice and reducing its effective bulk viscosity, as observed in Pine Island and Thwaites Glaciers where upstream fracturing correlates with flow acceleration (Lhermitte et al., 2020; Sun and Gudmundsson, 2023; Surawy-Stepney et al., 2023). This damage interacts with basal slip–where ice slides over bedrock–through stress redistribution that enhances basal crevassing (Bassis and Ma, 2015) and by facilitating meltwater penetration, which reduces basal friction and further accelerates slip (Sun et al., 2021; Clayton et al., 2022). Consequently, damage evolution is critical for projecting long-term ice flow changes and land ice stability (Bassis et al., 2024), though model uncertainties persist regarding damage parameters and feedback mechanisms.

To further our understanding of the glacier detachment mechanism, we study the 2018 Sedongpu glacier detachment in this paper. Firstly, we describe the environmental conditions of the study site. Then, we introduce the numerical model methods we used, where a novel ice stiffness-basal slip positive feedback coupling scheme is implemented, following by the results and discussions.

## 2  Study region

The study area (Sedongpu Glacier) is situated within the Namcha Barwa-Gyala Peri massif in the southeastern Tibetan Plateau (Fig. 1a), characterized by several distinctive features, including high tectonic activity, significant variations in topography and deep incisions caused by the Yarlung Tsangpo River. The Indian summer monsoon penetrates through the Yarlung Tsangpo Canyon, resulting in the longest annual rainy season on the Tibetan Plateau (Yang et al., 2013). In 2019–2020, the Medog County, located about 60 kilometers from the Sedongpu Valley, received over 1200 millimeters of precipitation, with 56.6% occurring from June to September and 32.4% in the spring season (March-May) (Li et al., 2022) .

As a result, the abundant monsoonal rainfall has led to the presence of 141 modern temperate glaciers in the Namcha Barwa-Gyala Peri region. Additionally, the accumulation of thick Quaternary glacial deposits (Montgomery et al., 2004), along with

these unique climatic, and topographic conditions have historically resulted in significant natural disasters and river blockages (Chen et al., 2020). The Sedongpu Glacier was underlain by a thick sediment/moraine layer which was eroded during the 2018 detachment event, forming a canyon up to 300 meters deep (Kääb et al., 2021; Kääb and Girod, 2023).

According to the Randolph Glacier Inventory (RGI) 6.0, the Sedongpu valley is home to five major glaciers. The largest of these is the Sedongpu Glacier (RGI60-13.01428), covering an area of 5.0 km$^2$, the majority of which detached in October

2018 (Kääb et al., 2021). The glacier surface is heavily covered with debris, while the underlying bedrock primarily consists of Proterozoic marble and gneiss (Chen et al., 2020).

## 3   Datasets

We generated two high-resolution digital elevation models (DEMs) using commercial stereo optical satellite images: a 1-meter-resolution SPOT6 image captured on November 13, 2015, and a 0.5-meter-resolution Pleiades-1A image captured on

December 30, 2018. These images were processed in PCI Geomatica software (Banff Service Pack 4) with the OrthoEngine module. The ice below 4300 m a.s.l. of Sedongpu Glacier was completely detached in October 2018 , exposing the underlying bed (Li et al., 2022; Kääb et al., 2021). Therefore, the December 2018 DEM represents the bed topography, and the November 2015 DEM is assumed to represent the surface topography. The final DEM difference products had a relative mean vertical accuracy of 1.3±3.2 m from November 2015 to December 2018 over stable flat ground inside the Sedongpu valley (Fig. 2a).

We estimated local ice thickness by calculating elevation differences between the pre-detachment glacier surface and the post-detachment exposed bed topography at locations where substantial ice detachment occurred. These values provided first-order estimates of ice thickness and were used as discrete constraints in the GlaTE software (Langhammer et al., 2019), which infers distributed ice thickness by optimally combining observational data with glaciological modeling in an inversion framework. The modeling component follows the method of (Clarke et al., 2013), which approximates basal shear stress as a

function of surface slope and apparent mass balance under a shallow-ice assumption. The inversion is formulated as a linear optimization problem with smoothness regularization, implemented via a smoothing matrix to enforce structural simplicity in the solution. We provided the estimated thickness points, a DEM, and the glacier outline as inputs to GlaTE. After obtaining the distributed ice thickness, we extracted the glacier geometry along the main centerline, which was generated following the method proposed by Kienholz et al. (2014). This flowline geometry was then used as input for the PoLIM simulations.

We generated a spatially distributed estimate of XY surface displacements by applying a Normalized Cross Correlation algorithm to two phases of 3-meter Planet Labs optical satellite data in daily resolution (5 June 2018 and 18 September 2018) using ImGRAFT (Messerli and Grinsted, 2015) (Fig. 2b). A search window of 10×10 pixels (30×30 m) was used to compute the magnitude and directions of the displacement vectors. Surface velocities greater than 400 cm/day were considered as noise and were filtered out, we interpolated the velocity values in the data gaps using cubic spline interpolation (Mishra et al.,

2022). The uncertainty of surface velocity was obtained by calculating the mean displacement (5.26 m; 5.01 cm/d) from the non-glacial test areas.

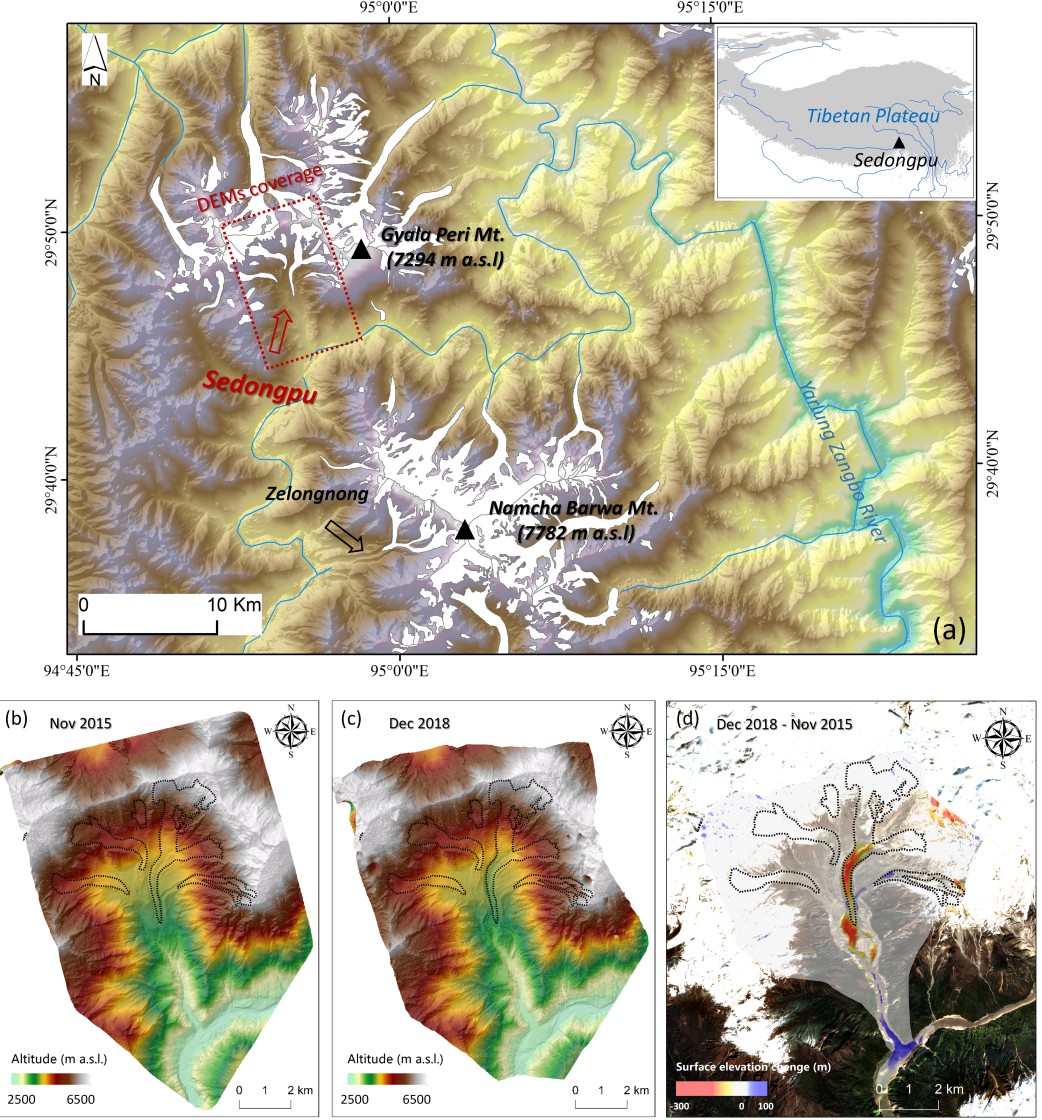

**Figure 1.** The location of Sedongpu valley and the glacier distribution around Namcha Barwa Mt and Gyala Peri Mt. (a). High-resolution DEMs generated from the stereo optical satellite images in November 2015 (b) and December 2018 (c) showing the surface and bed topography before and after the glacier detachment in 2018. The DEM difference between 2015 and 2018 can be seen in (d).

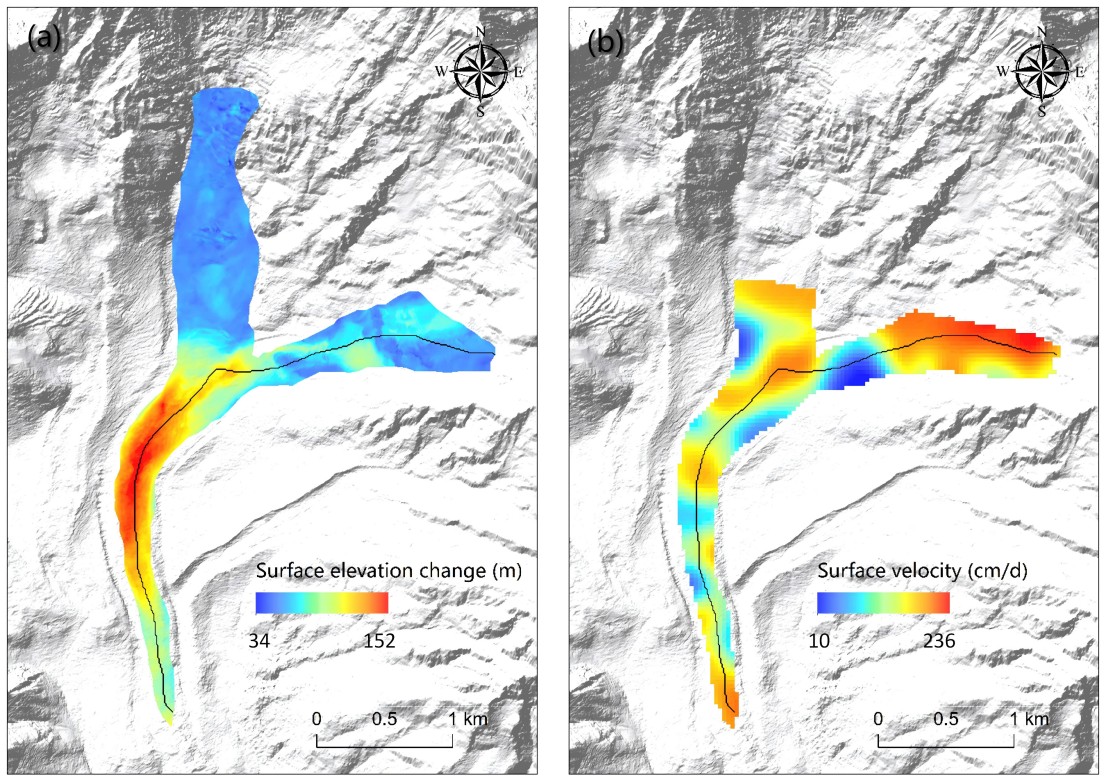

**Figure 2.** Ice surface elevation changes during 2015-2018 (a). Ice surface mean speed from June to September, 2018, prior to the ice detachment occurrence (b). The solid black curve represents the center flowline we use in this study.

## 4 Model descriptions

### 4.1 Ice flow model

In this study, we use a two-dimensional high-order ice flow model named as PoLIM (Polythermal Land Ice Model) (Zhang et al., 2013; Wang et al., 2018, 2020). PoLIM is developed according to the hydrostatic approximation, where the horizontal gradient of the vertical velocity is neglected in the viscous rheology and momentum equation (Blatter, 1995; Pattyn, 2002; Greve and Blatter, 2009). The momentum conservation equation of PoLIM is given by:

$$\frac{\partial}{\partial x}(2\tau_{xx} + \tau_{yy}) + \frac{\partial \tau_{xy}}{\partial y} + \frac{\partial \tau_{xz}}{\partial z} = \rho_i g \frac{\partial s}{\partial x}, \tag{1}$$

where $x$ represents the streamline direction, $y$ represents the transverse direction, and both $x$ and $y$ axes lie in the horizontal plane, while $z$ represents the vertical direction, and $s$ represents the surface elevation of the glacier. In addition, $\rho_i$ represents the ice density (set as a constant), and $g$ represents the acceleration due to gravity. Finally, $\tau_{ij}$ is the component of the deviatoric

stress tensor, which can be calculated from the strain rate

$$\tau_{ij} = 2\eta\dot{\epsilon}_{ij}, \tag{2}$$

where $\dot{\epsilon}_{ij}(i, j = 1, 2, 3)$ are the corresponding strain-rate components, and $\eta$ is the effective viscosity, calculated as:

$$\eta = \frac{1}{2}A^{-1/n}\dot{\epsilon}_e^{(1-n)/n}, \tag{3}$$

where $n$ is the flow law exponent and the effective strain rate $\dot{\epsilon}_e$ is defined as

$$\dot{\epsilon}_e^2 \simeq \left(\frac{\partial u}{\partial x}\right)^2 + \left(\frac{\partial v}{\partial y}\right)^2 + \frac{\partial u}{\partial x}\frac{\partial v}{\partial y} + \frac{1}{4}\left(\frac{\partial u}{\partial y}\right)^2 + \frac{1}{4}\left(\frac{\partial u}{\partial z}\right)^2. \tag{4}$$

Then the effective stress ($\sigma_e$) can be calculated by

$$\sigma_e = 2\eta\dot{\epsilon}_e. \tag{5}$$

Sedongpu Glacier is a typical maritime glacier in southeastern Tibet. In this study, we assume Sedongpu Glacier is temperate and set $A$ as a constant for ice temperature close to $0\ ^\circ$C (Cuffey and Paterson, 2010) (Table 1), i.e., we do not include a temperature solver in our model. At the glacier surface, we use a stress-free boundary condition, and at the glacier base, we apply a linear friction law prior to the occurrence of glacier detachment,

$$\tau_b = -\beta u_b, \tag{6}$$

where $\tau_b$ is the basal stress, $u_b$ is the basal sliding speed, and $\beta$ is the basal friction parameter. $\beta$ is hold constant in time before detachment. The glacier evolution is described by the mass continuity equation,

$$\frac{\partial H}{\partial t} = -\frac{\partial(\bar{u}H)}{\partial x} + m_s, \tag{7}$$

where $H$ is ice thickness, $t$ is model time, $\bar{u}$ is depth-averaged velocity, and $m_s$ is surface mass balance.

The model applies a finite difference discretization method. We set model time step to 0.5 second, and set $m_s$ to 0 given a very short model time span (25 minutes) in this study. We use a $\Delta x = 48$m in $x$ and 20 vertical layers in $z$ with a terrain-following coordinate. By these numerical settings, our model can satisfy the Courant–Friedrichs–Lewy (CFL) condition and keep numerical stability during forward runs. All model constants and parameters can be found in Table 1. Note that the values of critical strain and intact strength are from Bassis et al. (2021).

From the model descriptions above, we can see that glacier movement consists of two components: internal deformation and basal sliding. Internal deformation is influenced by ice viscosity. Decreased viscosity lead to increased ice flow. The basal sliding is controlled by the friction at ice-bed interface. Factors such as soft sediments and basal meltwater lubrication will reduce the basal friction, consequently accelerating ice flow, which are not considered separately but taken as a result of changing basal frictions in the sliding law in this study.

**Table 1.** Model parameters and constants used in our experiments

| Symbol | Description | Value | Units |
|--------|-------------|-------|-------|
| $\rho_i$ | ice density | 910 | kg m$^{-3}$ |
| $g$ | gravitational constant of acceleration | 9.81 | m s$^{-2}$ |
| $n$ | flow law exponent | 3 | |
| $A$ | rate factor | $3.17\times10^{-24}$ | Pa$^{-n}$ s$^{-1}$ |
| $\epsilon_c$ | critical strain | 0.1 | |
| $\tau_c$ | intact strength of ice | $5\times10^6$ | Pa |
| $p$ | step-size parameter | 1.5 | |
| $\Delta t$ | model time step | 1 | s |

### 4.1.1 Model initialization prior to detachment

Before simulating the Sedongpu detachment processes, we need to initialize the ice flow model using observed ice surface velocity data. Following Arthern and Gudmundsson (2010) and Arthern et al. (2015), we solve the basal friction coefficient using the Robin inversion algorithm by iteratively minimizing the cost function $J$ across the basal domain

$$J = \int_{\Gamma_b} \beta \left| \boldsymbol{u}^D - \boldsymbol{u}^N \right|^2 \, \mathrm{d}S, \tag{8}$$

where $\boldsymbol{u}^D$ is the observed ice surface velocity and $\boldsymbol{u}^N$ is the ice surface velocity solved in the model by applying a stress-free

surface boundary condition. This cost function represents the mismatch between the Neumann and Dirichlet velocity fields. Following Arthern et al. (2015), the basal friction coefficient was updated as follows,

$$\beta_{i+1}(x,y) = \beta_i(x,y) + \alpha_\beta \left( \left| \mathbf{u}_b^N \right|^2 - \left| \mathbf{u}_b^D \right|^2 \right), \tag{9}$$

where $\beta_i$ is the basal friction at the $i$-th iteration step, $\mathbf{u}_b^N$ and $\mathbf{u}_b^D$ are basal velocity for Neumann and Dirichlet iterations, $\alpha_\beta$ is a positive parameter that determines the step size, given as

$$\alpha_\beta = \frac{\beta_n}{\left| \mathbf{u}_b^D \right|^p} \frac{\left( \left| \mathbf{u}_b^N \right|^p - \left| \mathbf{u}_b^D \right|^p \right)}{\left( \left| \mathbf{u}_b^N \right|^2 - \left| \mathbf{u}_b^D \right|^2 \right)}, \tag{10}$$

where $p$ is a positive parameter, as given in Table 1.

### 4.2 Yield strength and basal slip coupling scheme

The overall model framework can be seen in Figure 3. In order to simulate the tipping mechanism of Sedongpu detachment, we implement a numerical scheme that couples basal slip and ice stiffness, following the approach in Bassis et al. (2021), which

integrates the continuum and discrete processes of ice flow. In this scheme, the new ice viscosity is calculated as

$$\eta_{\text{new}} = \eta_{\text{min}} + \left[ \frac{1}{\eta} + \frac{1}{\eta_{\text{diff}}} + \frac{1}{\eta_{\text{plas}}} \right]^{-1}, \tag{11}$$

where $\eta$ is the viscosity for Glen's power law creep flow (Eqn 3), $\eta_{\text{diff}}$ and $\eta_{\text{plas}}$ are the viscosity for diffusion creep and plastic deformation when ice failure occurs (see the supplementary materials in Bassis et al. (2021)), respectively. $\eta_{\text{min}}$ is a tunable minimum viscosity for numerical stability. The inclusion of the viscosity $\eta_{\text{plas}}$ in the fracture process indicates that stress does

not increase with increasing strain when the ice mass reaches the yield stress.

The presence of a "plastic" viscosity results in a low stiffness value and high velocity, as the ice viscosity remains relatively low. This phase corresponds to the development of ice crevasses and prevalent ice failure in reality. Additionally, the yield strength is not stable and constant; it decreases with the acceleration of ice flow, resulting in an unstable ice flow pattern (Bassis et al., 2021),

$$\tau_y = \max \left\{ \tau_c - (\tau_c - \tau_{\text{min}}) \frac{\epsilon_p}{\epsilon_c}, \tau_{\text{min}} \right\}, \tag{12}$$

where $\tau_y$ is the yield strength of ice, $\tau_c$ is the intact strength (see Table 1), $\tau_{\text{min}}$ is a tunable, prescribed minimum yield stress, $\epsilon_c$ is the critical strain, and $\epsilon_p$ is the plastic strain accumulated in faults and fractures which can be calculated during the model run. For the basal sliding law, we also consider the impact of yield strength of basal ice, similar to Bassis et al. (2021),

$$\tau_b = - \left[ \frac{1}{\beta} + \frac{u_b}{\tau_y} \right]^{-1} u_b, \tag{13}$$

where $\tau_b$ and $u_b$ are basal stress and speed, respectively, and $\beta$ is the basal friction we use before the detachment. Here $\tau_y$ is the ice yield stress at the bed, representing interactions between basal ice and soft till, i.e., basal ice stress will continue to decrease if it exceeds yield strength. With this improvement, basal slip is enhanced as ice failure increases and basal ice strength decreases. Therefore, this mechanism can better capture the dynamics of the soft and thick till layer underneath Sedongpu glacier, which could deform significantly during detachment (Yang et al., 2023).

In Figure 3, we present a diagram of our numerical modeling processes. In traditional ice flow models (e.g., Glen's flow law), increased stress enhances viscosity during the elastic deformation phase, promoting slow and stable ice motion. However, basal sliding formulations (e.g., the Weertman sliding law) relate basal friction to velocity and stress while neglecting ice stiffness. This limitation hinders the simulation of ice detachment processes.

The proposed model framework incorporates the plastic phase of ice flow by introducing a yield stress tipping point, estab-

lishing two positive feedback mechanisms: (1) within internal ice deformation and (2) at the ice-bed interface. We first initialize the ice flow model and compute stresses during forward simulations. These stresses are then compared to the ice yield strength. When stress reaches this threshold, ice stiffness and yield strength decrease, triggering further yielding and increasing glacier vulnerability. Then, reduced basal ice stiffness lowers basal friction, facilitating accelerated sliding. This dual process creates a positive feedback mechanism coupling ice stiffness and basal slip.

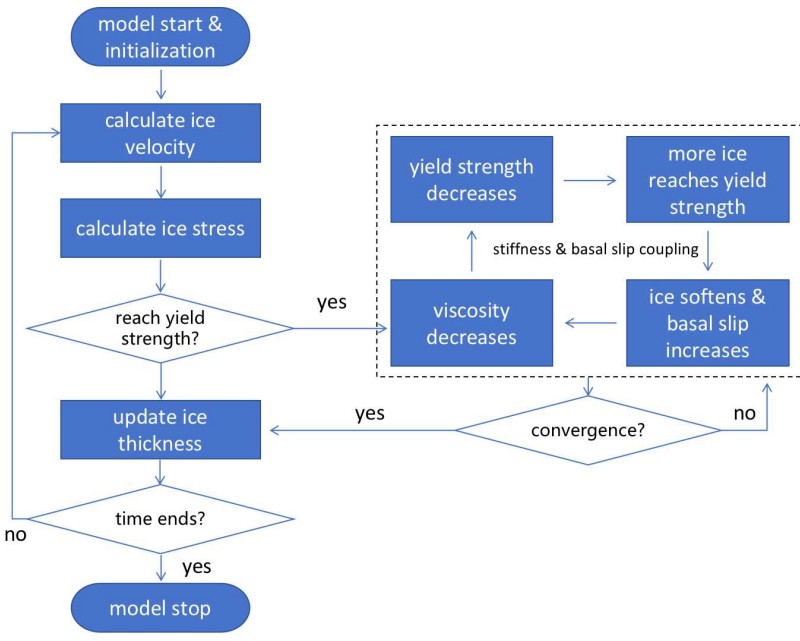

**Figure 3.** The flow diagram of the numerical modeling procedures in this study.

## 5 Model results

### 5.1 Sedongpu detachment simulation

As shown in Figure 4, we inverted for the basal sliding coefficient of Sedongpu Glacier. The 2018 Sedongpu Glacier detachment occurred on October 17. Prior to this event, several ice-rock avalanches in the glacier's upper region between June 2014 and October 2017 increased flow velocity from around 0.3 m/d in 2017 to 25 m/d by September 2018 (Kääb et al., 2021). Due to insufficient high-resolution observations, it is difficult to accurately simulate the glacier's acceleration during this period. Therefore, to model the 2018 detachment, we initialize our simulation using the 2015–2018 mean observed ice velocity preceding the event. The results show faster flow in upstream regions with lower basal friction (likely due to steeper slopes), while downstream regions exhibit slower motion where greater ice thickness corresponds to higher basal friction.

Environmental forcings may have acted as external triggers for the 2018 Sedongpu detachment. From January to October 2018, the region experienced a historical mean temperature increase rate of 0.039 K yr$^{-1}$ (Liu et al., 2019). Although precipitation during this period was below historical observations, intense rainfall occurred 2–4 days before the detachment event. This rainfall likely softened basal ice and accelerated flow, altering internal ice dynamics and ultimately triggering the abrupt collapse.

As shown in Figure 5, our simulation successfully reproduces the decrease in ice thickness and increase in ice velocity associated with a glacier detachment. We activated the yield strength and stiffness-slip coupling mechanism at $t_0 = 5$ minutes after confirming model stability pre-detachment, then ran the simulation for 25 minutes. Within several time steps, the bulk

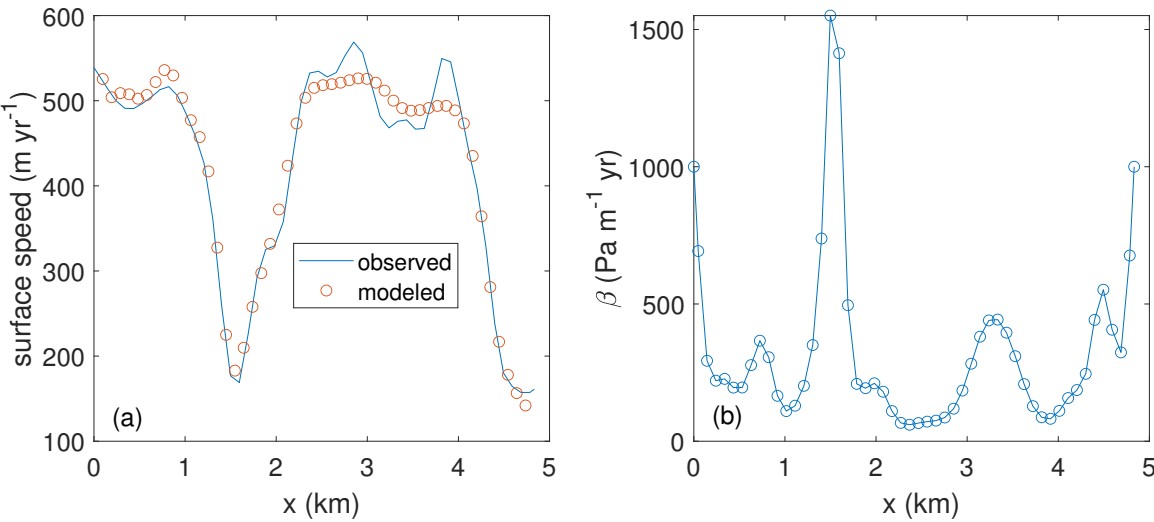

**Figure 4.** The The comparison of observed and modeled surface speed after inversion (a), and the inverted basal sliding parameter using the Robin inversion algorithm (b). The inversion is based on velocities observed by remote sensing from 2015 to 2018.

viscosity of the glacier decreased to its minimum ($\eta_{\min}$), significantly enhancing ice fluidity. Consequently, ice velocity surged from less than 1 m per hour to around 90,000 m per hour, with a mean speed of approximately 34,000 m per hour over the 25-minute simulation period. As the glacier rapidly thinned due to mass loss, velocity stabilized at lower values, reaching a
new steady state.

The mean effective stress of the Sedongpu Glacier drastically increases from 200 kPa prior to detachment to around 1000 kPa as ice flow accelerates after the detachment starts. Generally, the changes in englacial stress follow a similar pattern to that of ice speed. During the model run, the glacier mass is rapidly transported from upstream to downstream, resulting in a reduction in glacier thickness by approximately 80% within only 6.3 minutes and 90% within 11 minutes after $t_0$ (Figure
5). Note that the oscillation of speed and effective stress after collapse is related to the prescribed minimum ice thickness (1 m) in our simulations: this unnatural model setting can produce non-smooth distributions in ice thickness, and thus speed and stress, correspondingly. Our model results closely match a previous estimation from October 17, 2018: the total detachment time lasted around 6.7 minutes and the mean ice speed was approximately 20 m s$^{-1}$ (72,000 m per hour) (Liu et al., 2019). In fact, according to the seismic signal data in a previous report (Yao et al., 2022), the 2018 Sedongpu detachment lasted around
5 minutes, which provides strong evidence for validation of our model results.

The mean effective stress in Sedongpu Glacier increased drastically from 200 kPa pre-detachment to around 1000 kPa during detachment acceleration. Generally, higher ice flow velocities result in greater englacial stresses, increasing the vulnerability of ice regions to detachment instability. During the simulation, rapid mass transfer from upstream to downstream reduced glacier thickness by  80% within 6.3 minutes and 90% within 11 minutes after $t_0$ (Figure 5). Post-collapse oscillations in velocity
and stress arise from our prescribed minimum ice thickness (1 m) and the internal numerical instability, an artificial constraint

that creates non-smooth thickness distributions and corresponding fluctuations. Our modeled detachment duration (6.7 min) and mean velocity (approximately 20 m s$^{-1}$, i.e., 72,000 m per hour) align closely with estimates for the 17 October 2018 event from a previous study (Liu et al., 2019). This validation is further strengthened by seismic data indicating a 5-minute detachment duration (Yao and An, 2022).

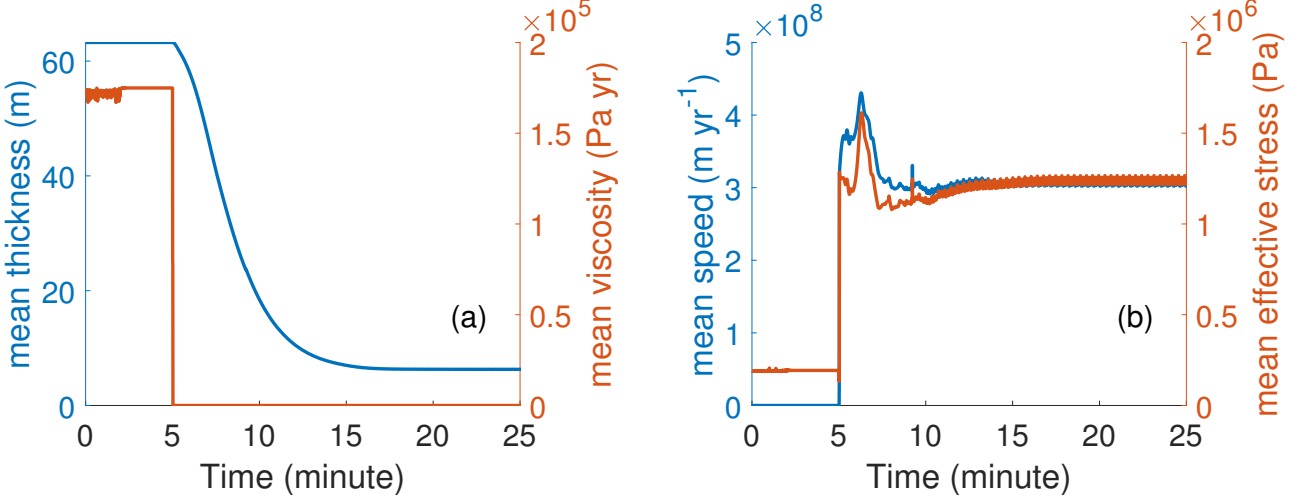

**Figure 5.** The transient changes of mean ice thickness, viscosity (a), ice speed and effective stress (b) in time for the Sedongpu glacier during a 25-minute model run. The detachment begins at minute 5. All variables here are shown in their normalized form in time.

### 5.2 Model sensitivity

In this study, we assume a spatially uniform yield strength at model initialization (hereafter defined as "initial yield strength") across Sedongpu Glacier. The initial yield strength is a crucial parameter that determines abrupt glacier detachment. As it typically varies between 100–1000 kPa (Cuffey and Paterson, 2010), we conducted sensitivity experiments to assess its impact on glacier dynamics. This allows us to estimate the destabilizing threshold (tipping point) for Sedongpu Glacier. Figure 6a shows cases with initial yield strengths between 300 and 500 kPa. Detachment occurs only when the initial yield strength is set below 440 kPa, indicating that the mechanical properties of glacier ice control sudden collapse when failure exceeds critical thresholds. While subglacial hydrology-ice dynamics coupling can explain glacier surge mechanisms (Thøgersen et al., 2019), it cannot reproduce extreme detachment instability without accounting for dramatic ice weakening. This is demonstrated by cases with initial yield strengths above 450 kPa in Figure 6a, where glacier ice remains intact throughout the simulation.

Additionally, the rate of ice loss can be significantly influenced by two tuning parameters, $\tau_{\min}$ (prescribed minimum yield stress) and $\eta_{\min}$ (prescibed minimum viscosity) (Equations 11 and 12), the choice of model parameters, in our model. These parameters determine the maximum ice flow fluidity and thus can greatly impact the rate of mass loss during detachment. As shown in Figure 6b, for the same initial yield stress value (300 kPa), a combination of lower minimum stress ($\tau_{\min}$) and minimum viscosity ($\eta_{\min}$) values leads to a greater reduction in glacier mass. However, this does not change the tipping point

of glacier collapse, which is determined by ice flow dynamics and mechanical properties. Once some ice regions yield, the yield strength further decreases, making the glacier even more vulnerable to fracturing. Consequently, this plastic, accelerating ice flow quickly affects the entire glacier, leading to drastic detachment.

We should note that the yield strength of glacier ice is generally an unknown parameter and is highly heterogeneous in space. In this study, we approximate this tipping threshold value as a spatially uniform constant for the convenience of explaining the 230 critical role ice strength plays in glacier detachment. The positive feedback between the weakening of ice stress beyond yield strength and the rate-weakening basal friction likely contributed together to the collapse of the Sedongpu Glacier in 2018.

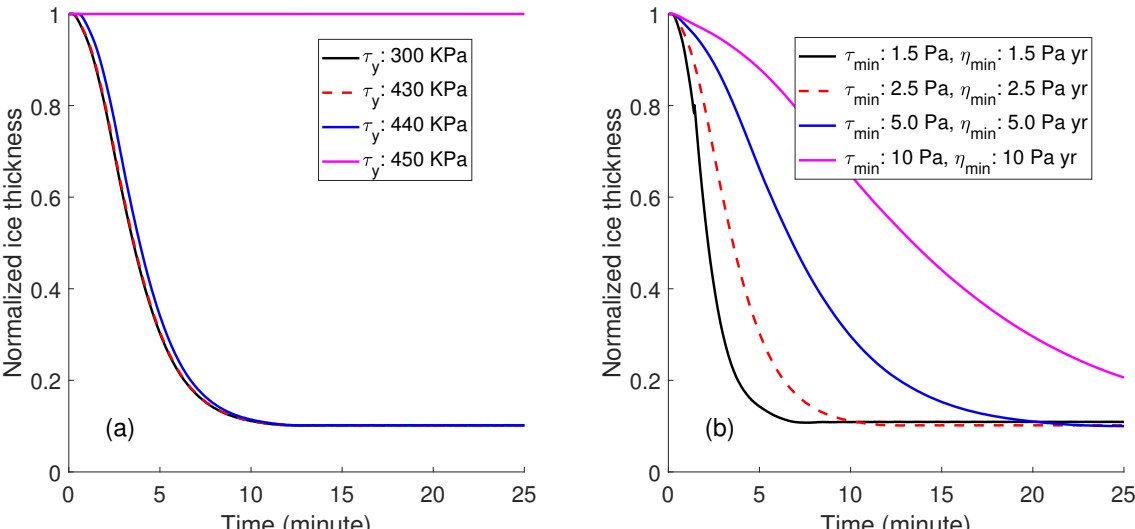

**Figure 6.** The sensitivity of mean ice thickness changes of Sedongpu to different initial yield stress values (a), model parameters $\tau_{\min}$ and $\eta_{\min}$ (b) during the 25 minute model time span. The detachment begins at minute 5. The $y$-axis indicates the normalized value of mean ice thickness ($\bar{H}_t / \bar{H}_{t0}$) along the flowline.

### 5.3 The tipping processes of Sedongpu detachment

We can gain further insight into the detachment mechanism from Figure 7, which shows the dynamic changes of Sedongpu Glacier at three subsequent time steps after activating the stiffness-slip coupling mechanism at time $t_0$. When the initial yield 235 strength is set to 300 kPa, the ice stress exceeds the yield strength near the glacier's head, terminus, and around km 1.3 at $t_0$. By the next time step, the ice stress across the entire glacier surpasses the yield strength, resulting in whole plastic deformation and a significant acceleration of ice flow.

However, when the initial yield strength is increased to 430 kPa and 500 kPa, the plastic deformation regions near the head and terminus disappear after updating the velocity and ice thickness in the following two time steps. At 430 kPa, the entire

glacier is in plastic deformation by the third time step, and the ice velocity surges dramatically to nearly 100,000 m per hour, indicating glacier detachment. In contrast, at 500 kPa, the glacier remains stable over the next three time steps.

Previous studies by Kääb et al. (2018) and Gilbert et al. (2020) conducted in-depth analyses of the 2016 Aru Glacier collapse, revealing that the catastrophic event was controlled by multiple factors, including a deformable substrate, increased driving stress, temperate ice conditions, and connections to subglacial water. The transition from slow ice movement to catastrophic instability may have developed over months or even years before the collapse. In this study, we focus on the dynamic instability during abrupt glacier detachment—a process that can occur within minutes. We argue that the transition from elastic to plastic ice deformation likely plays a non-negligible role in this rapid phase. Once ice stress exceeds the yield strength, a tipping point is reached, triggering localized acceleration that abruptly propagates across the entire glacier. This leads to highly fractured ice geometry within seconds to minutes. Such a mechanism creates a positive feedback mechanism of accelerating ice flow and strength reduction—a process traditionally not fully considered in numerical glacier flow simulations.

This glacier detachment tipping mechanism may also apply to other high-risk glaciers surrounding the Sedongpu Valley. For example, the Zelongnong Glacier (Fig. 1a) represents a potential candidate for future detachment events. By assuming dynamic characteristics similar to Sedongpu (e.g., comparable initial yield strength), we could assess Zelongnong's detachment risk through surface velocity monitoring and numerical simulations of its internal stress regime. Implementing such an approach would require detailed investigations of glacier geometry to establish a reliable ice flow model and develop an effective early warning system.

## 5.4 Discussion

Similar to Kääb et al. (2018), our modeling approach is still based on the Glen's flow law framework but incorporates a novel stiffness-basal slip coupling scheme. While we assume constant ice density during detachment, this simplification may limit the model's ability to fully capture the dynamics of Sedongpu Glacier's highly fractured detachment. The continuum modeling scheme we employ (Bassis et al., 2021) simultaneously accounts for ice flow and failure, though it cannot fully replicate discrete methods in simulating ice collapse processes. In addition, our bed elevation extraction–based on pre- and post-detachment geometry comparisons–introduces potential uncertainties. These arise from the soft basal sediment characteristics of Sedongpu Glacier (Kääb et al., 2021) and possible subglacial geometry alterations during detachment.

Furthermore, our model does not incorporate thermal coupling or basal hydrology schemes, potentially neglecting key physical mechanisms involved in glacier detachment. For instance, Thøgersen et al. (2019) identified a velocity-strengthening-weakening transition that governs surge initiation, though their framework assumes intact ice and slow movement–conditions that may not adequately capture rapid detachment dynamics. While we acknowledge the importance of ice-bed interactions with basal hydrology (Hoffman and Price, 2014), our current implementation employs a simplified sliding law (Eqn 13) to couple basal till strength with ice flow. This approach, though computationally efficient, could be enhanced in future work to better represent these complex processes.

Kääb et al. (2021) analyzed the force balance of simplified, slab geometries and marked Sedongpu Glacier as instable. In fact, for stable glaciers with basal cavities, basal drag is constrained by an upper limit known as Iken's bound (Helanow et al.,

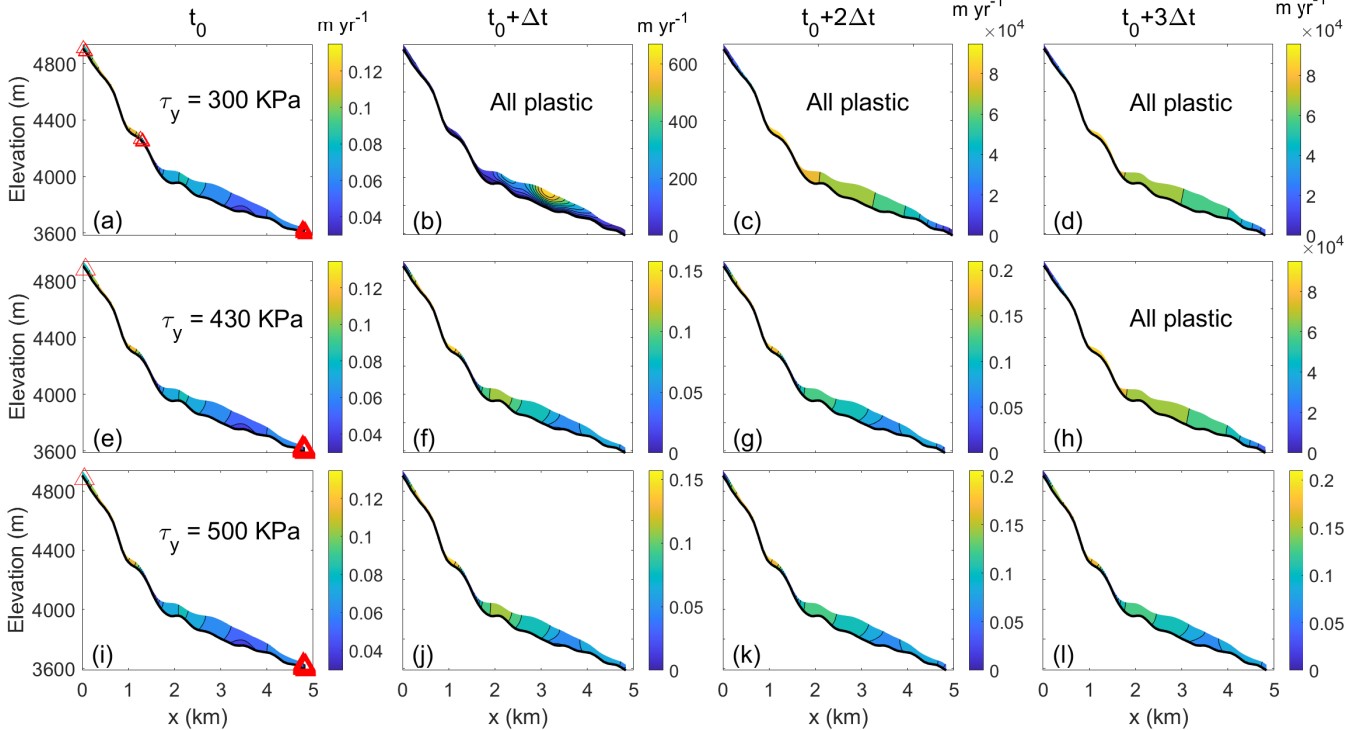

**Figure 7.** The changes of Sedongpu glacier velocity at 4 time steps after the ice stiffness-basal slip coupling mechanism is triggered. (a, b, c, d) initial yield strength is set as 300 kPa; (e, f, g, h) initial yield strength is set as 430 kPa; (i, j, k, l) initial yield strength is set as 500 kPa. The red triangles indicate the locations where ice stress exceeds yield strength. The panels with "All plastic" indicate that the entire glacier is across the tipping points of yield strength and having plastic deformation with a drastic speed acceleration. The colorbars show the ice speed of Sedongpu Glacier with the unit of m yr$^{-1}$. The rapid transition to plastic flow occurs for low initial yield strengths.

2020, 2021),

$$\tau_b/N \leqslant m_{\max},$$   (14)

where $\tau_b$ is the basal shear stress, $N$ is the effective pressure and $m_{\max}$ is the maximum value of the up-glacier-facing slopes of obstacles. Here we assume $N$ is the overburden ice pressure (no basal water pressure) and set $m_{\max}$ to the maximum bed slope. Figure 8 shows that once the detachment instability mechanism is triggered at $t_0$, the ratio $\tau_b/(Nm_{\max})$ in the upstream region of Sedongpu Glacier rapidly exceeds 1 (violating Iken's bound), which aligns closely well with the timing of the detachment
event.

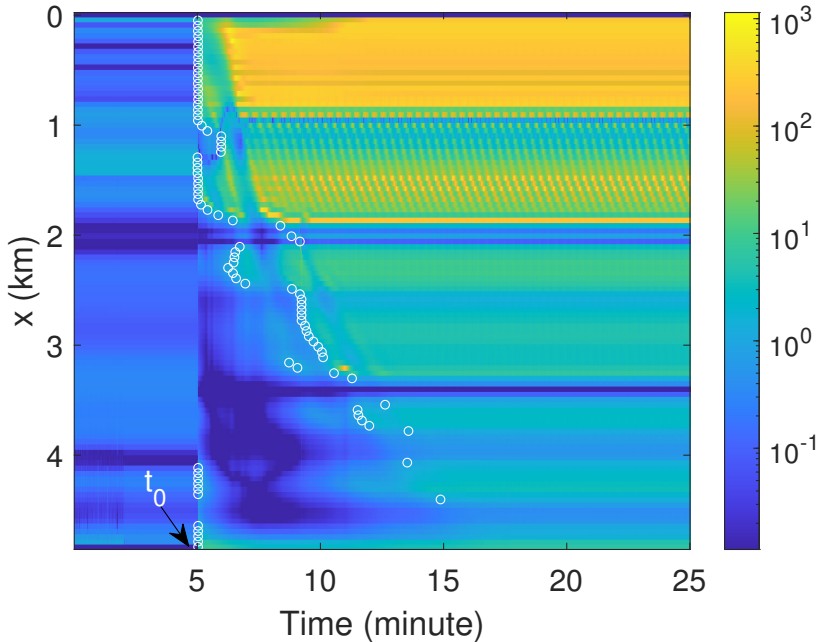

**Figure 8.** The spatio-temporal changes of the ratio $\tau_b/(Nm_{\max})$ (unitless). The detachment instability mechanism is triggered at $t_0$ (5 minutes). The white circles mark the initial occurrence of $\tau_b/(Nm_{\max}) > 1$ along $x$ after detachment begins.

## 6 Conclusions

Glacier detachment results from complex interactions among multiple factors, including crevasse formation, meltwater infiltration, subglacial hydrology, and basal slip properties. However, observational data limitations and current modeling constraints make it difficult to fully incorporate all these elements in numerical simulations. This study focuses on investigating key triggering mechanisms for glacier detachment, with particular emphasis on two critical aspects, including the transition process of internal stress evolution within the glacier body, and the characteristic behavior of basal sliding.

Our findings demonstrate that the Sedongpu Glacier's transition from slow deformation to abrupt collapse can be captured by analyzing evolving internal stress states. By incorporating critical controlling factors–particularly ice yield strength–into ice flow models, we can identify potential early warning signals of detachment. From the model results, we find that glacier ice detachment occurs when the initial yield strength drops to approximately 430 kPa, indicating that the ice's mechanical properties are critical in triggering abrupt collapse when mechanical stress exceeds critical failure thresholds.

To advance this study, future efforts should extend the current two-dimensional model to three dimensions and further investigate the relationships between Iken's bound of the glacier bed and basal sliding/hydrology. By developing a more advanced numerical modeling approach, we can study glaciers in neighboring regions and estimate changes in their surface and internal ice stresses using in-situ and remote sensing observations. Subsequently, we could assess the detachment risk of surrounding

glaciers, or even on a larger scale, by examining the relationships among ice stress, yield strength, surface landforms (e.g., crevasses), and basal sliding features.

*Code availability.* The ice flow model PoLIM can be freely accessed at https://github.com/WangYuzhe/PoLIM-Polythermal-Land-Ice-Model. The version for simulating Sedongpu Glacier detachment can be found at https://zenodo.org/records/15831881.

*Data availability.* The velocity and boundary data of Sedongpu Glacier can be obtained from https://zenodo.org/records/15831881.

*Author contributions.* TZ and WY conceived this study. TZ designed and constructed all model experiments. WY, YW, CZ and QL helped preparing the data of Sedongpu glacier. All authors contributed to the writing of the paper.

*Competing interests.* The authors declare no competing interests of this paper.

*Acknowledgements.* This work was supported by the National Key Research and Development Program of China (grant no. 2023YFF0805200),
Open Research Fund of TPESER (Grant No. TPESER202203), the National Natural Science Foundation of China grant 42271133 and 42271134, the Beijing Normal University Talent Introduction Project of China (12807-312232101), Basic Research Fund of CAMS (2023Z004), and Science and Technology Projects in the Tibet Autonomous Region (grant no. XZ202401ZY0003). We thank two anonymous reviewers and the editor (Gong Cheng) for their great helps in improving this manuscript.

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
