# Peer review of "Numerical Modeling of Ice Detachment Tipping Processes: Insights from the Sedongpu Glacier, Southeastern Tibetan Plateau"

_EGUsphere, 2025_

## Referee Comment (RC1)

Zhang et al. 2025 "Numerical modeling of the tipping processes of ice detachment: a case study of Sedongpu Glacier in the Southeastern Tibetan Plateau"
Reviewer comments

General comments

The authors present a study of glacier detachment at Sedongpu Glacier from a numerical modeling perspective that attempts to recreate conditions of glacier attachment by simulating changes in glacier viscosity and feedbacks with basal sliding. The topic and motivation is really important, as glacier detachments deserve more scientific attention and can be destructive to downstream communities and infrastructure (as the authors note as well). I think the topic of the manuscript in itself has a lot of merit, and it's cool that the authors have been able to reproduce catastrophic collapse of the glacier. Because of this, I think that this paper is a good candidate for publication. However, I have serious concerns about the approach the authors take as well as the structure and presentation of the manuscript. These can be categorized in the following areas.

(1) I am somewhat unconvinced about the authors' choice to model glacier detachment from the perspective of a rate-weakening basal slip and variable ice viscosity. The authors claim that the feedback between ice stiffness and basal slip plays a major role in the instability that causes the glacier to detach within minutes in their simulations. However, there is very little discussion of how this feedback mechanism or its numerical implementation works. Neither have the authors cited any works that discuss this feedback. I remain unconvinced as to why this mechanism is responsible for the detachment both from a conceptual perspective and from their model results. The authors need to strengthen their explanation of this feedback and how/why they are modeling it. This could include adding a literature review section about damage mechanics and more explanation about why they decided to structure their coupled model the way they did.

(2) I am somewhat skeptical that the instability modeled in this manuscript (the viscosity-basal slip coupling) is truly a physical instability and not a numerical instability caused by the choice of model parameters. For example, when the coupling mechanism is activated, the viscosity of the ice goes from 60 to 0 Pa yr, which then triggers the rest of the glacier collapse. A more rigorous stability analysis is needed to ensure that this is a real physical phenomenon.

(3) The authors should provide some sort of validation or provide insights into the plausibility of their results. They repeatedly claim that the viscosity-basal slip feedback is more important than subglacial hydrology or till failure, but they do not provide any evidence to support this, either in the form of citing previous works or providing any sort of validation for their model. In addition, the authors do not show the misfit for their basal friction inversion, so we don't know whether or not their sliding parametrization is able to reproduce realistic glacier velocities.I think the authors should either include this

evidence or scale back their claim that this feedback is more important than other mechanisms which have been more extensively proven to play a role in detachments and surges (e.g., subglacial hydrology and geothermal heat).

(4) The authors don't include any sort of hydrology (subglacial or surface) in their modeling even though surface meltwater-driven hydrofracture and subglacial hydrology play a strong role in bulk ice toughness and basal slip. I get that this would be difficult to include in a model like this, but it would add a lot to the manuscript if the authors could at least address this in the text. They also talk about effective stress but have not defined it any of their model equations, and claim that high effective stress corresponds to ice acceleration which goes against the traditional glaciology definition of effective stress.

(5) I would like a lot more information on the implementation of the model or finite element scheme. What mesh resolution is used, what meshing software was used, and how the spatial and temporal resolution was determined. In addition, there should be more discussion of the implicit assumptions that go into the model and what effects these have on the results.

(6) The manuscript doesn't follow as much of a logical flow as I would prefer. There is insufficient explanation of the model and its assumptions at the beginning of the manuscript before the equations are presented. I go more in detail about this in the specific comments below.

(7) I also believe it could be helpful for a native English speaker to read through the next iteration of the manuscript as the language is often imprecise and not entirely scientific.

Specific comments
L5: "Yield strength of the glacier" is very general. As the authors mentioned later in the manuscript, the yield strength of the glacier is extremely heterogeneous. I would recommend changing this to something like "initial yield strength of glacier ice".

L8: "just several model time steps" is pretty vague

L9-10: How would these results be used for early warnings? Maybe you can talk about this later in the discussion.

L16: Merge this with the previous paragraph

L22: The "largest ever recorded event" of what? Of surges, of a glacier detachment? The language could be a bit more precise.

L24: The authors mention that remote sensing advances allow us to detect changes in surface features like crevasses and fractures but the authors did not use any of these data to validate the model results

L38: This is the first time the authors mention the ice stiffness-basal slip positive feedback. It would be great to have an additional few paragraphs explaining this feedback and motivating why the authors choose to model it. It would also be great to see some previous examples where this mechanism was suggested or modeled successfully before.

L41: Specify the glacier name again

L44: What is Medog? Is it a village, a weather station?

L48: Some information about the quality/texture of the Quaternary deposits would be useful here. Is it soft sediment or more coarse till? Also, how thick is the debris cover on the glacier?

Figure 1: (a) It would be great to highlight the glacier that is being modeled. (b) and (c ) the text is a little blurry - would be better to have higher resolution. Please also show the outline of the glacier in (b) and (c ) so we can get a better idea of where the glacier is inside the valley. Plotting the difference between the two DEMs would also probably be more useful (i.e., 2018 DEM minus 2015 DEM).

L56: I think this whole section (ice flow model) should go after Figure 2.

L57: What is the dimensionality of this model? It would be good to specify.

L57: PoLIM is a thermomechanical model. What are the thermal parameters specified here? What assumptions are made about heat flow? Is there any geothermal heat flux?

L70: Reference to Table 1 would be useful here.

L71: Justify why you are using a Weertman-type sliding law here, and why effective pressure (N) isn't important.

L83: There is also another model initialization section in the "results and discussion" (Section 5.1). Consider merging these two?

Table 1: What is the geothermal heat flux? Also, consider converting the rate factor A to Pa^-n s^-1 for easier comparison with other papers. It would also be good to include citations for the choices of critical strain and intact strength of the ice.

L90: Where do you get the Dirichlet and Neumann velocities from? What does it mean for velocity fields to be Neumann/Dirichlet? Also, what is the misfit from this inversion, and is the model able to reproduce realistic velocities?

L101: There needs to be an explanation of how you obtain eta_1 and eta_2 and what each of the three viscosities mean. Also, how is eta_min determined and what is the model's sensitivity to choice of eta_min?

L106: When and how does the viscosity transition to plastic viscosity?

L108: It would be helpful to cite previous works that have demonstrated good results with a model like this

L113: Are you using the linear Weertman-type or this rate limited one?

L117: Include a reference to Figure 2 before presenting the model equations. It would also be great to expand more on Figure 2 in the text.

Equation 13: Why is the ice yield strength in the expression for basal shear stress? In Bassis et al.'s supplement, the second term inside the brackets has tau_c (the intact strength) in the denominator, but here you have tau_y (yield strength).

L126: The "Datasets" section should be before the model equations.

L138: Need more information on what the "glaciological modeling results" are. Are they using a flow approximation? If so, what is being used? Some more details on how the glacier geometry was determined would be helpful here.

L139: What are some limitations to using a flowline model as you are doing, compared to a 2D model? How would this change the results?

L144: Why were higher velocities filtered out? It seems like a threshold of 400 cm/day could exclude important data if the glacier was moving faster.

Figure 3: Move the panel labels (a and b) to above the figures so they are more visible. Including a hillshade contour around the glacier domain would also be useful. In Panel A, you should use a diverging colormap so it's easier to see where the glacier thickness increased and decreased. Also, why is there data missing in the upper tributary in panel B?

L148: There was already a model initialization section. This should be merged with that. It doesn't make sense to have a model initialization in the Results/Discussion section.

L153: Why did you use the mean velocity from 2015-2018 instead of the velocities just prior to the detachment? Does Sedongpu exhibit seasonal variations, and how would that affect your calculation of the friction coefficient at the moment of detachment?

Figure 4: What is the resolution of this flow line model? How do these velocities compare to observations?

L159: The authors claim that external environmental forcings could be responsible for the detachment, but these forcings are not included in the model.

L164: Why is the coupling mechanism activated at t=5 minutes? What physical event does this represent? Does the model reach any sort of equilibrium prior to $t_0$? And why does the glacier instantaneously collapse when you activate the coupling mechanism - can you be sure that it isn't a numerical instability?

L167: 90,000 m/hour seems unrealistic. Can you comment on why it's so high, and what this could be representing?

L170: This is the first mention of effective stress in the entire paper. There is no mathematical or conceptual definition of effective pressure anywhere prior to this, and effective stress was not a parameter in any of the model equations shown. How is effective stress being calculated?

L171: An increase in effective stress should not result in ice acceleration, unless you are defining effective stress differently from the traditional glaciological definition. Higher effective stresses are associated with higher frictional contact between the ice and bed, and so a higher effective stress should correspond with the glacier slowing down. Effective stress needs to be defined explicitly in the ice flow model section and maybe again here.

L173: Is the 6.3 minutes from the beginning of the simulation or 6.3 minutes after $t_0$ the activation of the coupling mechanism?

Figure 5: It would be great if the authors could comment on the sudden change in viscosity from 60 to 0 Pa yr. It seems unrealistic to me that the viscosity would go to 0 within less than a minute.

L177: Nice relating with previous work

L184: If yield strength varies from 100-1000 kPa, why did you only test between 300-500 kPa?

L190: What is the sensitivity of the model to A and to the temperature of the ice?

L199: It would be great to include a conceptual explanation or discussion of why the coupling feedback mechanism almost instantaneously results in failure and rapid acceleration of the ice. This is the crux of the manuscript and should be explained in more detail.

L200: Earlier the authors suggested that hydrology doesn't play as much of a role as the viscosity-slip feedback mechanism. It is unclear to me then why rainfall events should be monitored as opposed to crevassing or softening of the glacier ice.

Figure 6: What is the normalized value? I don't think the normalization was defined clearly in the text or in the caption for the figures. Is it a spatial average? Maybe I missed it, but I don't understand the y-axis for these plots.

L206: What does ice becoming "yielded" mean?

L218-220: This sentence would make more sense if it were in the introduction. Or, if the authors wish to claim that the elastic-plastic transition is important, it would be helpful to include a scatter plot of stress vs. strain rate or something like that to show the glacier viscosity transitioning from elastic to plastic.

L224-225: "This has remarkable scientific and engineering applications for large infrastructures in the local regions" - I think it would be good to be more specific about which aspects of your results would be helpful for science/engineering applications. Otherwise, this sentence may make more sense if it were in the introduction or conclusion.

Figure 7: The panel labels are hard to see - maybe put them on top of each plot instead of at the bottom. Also the colorbars need units.

L226: This section should go closer to the beginning when you are introducing the model - it would be helpful to understand more of the model limitations from the outset.

L228: The assumption of constant ice density wasn't included in the model description at the beginning. It would be helpful to mention this in the "ice flow model" section.

L237: Could you write briefly about how including some of these mechanisms could affect the results? E.g., if you used Schoof's sliding law or Iken's bound, how would that change the results?

L248-249: What are the early warning signals of glacier detachment? These weren't discussed at all in the manuscript.

Technical corrections and typos
L38: "novel" -> "novel", "postive" -> "positive"
L101: "Glen's pow law" -> "Glen's power law"
L89: change "According" to "Following"
L163: "simulated" -> "simulate"
L183: "destablizing" -> "destabilizing"
L230: "descrete" -> "discrete"
L234: "hydrolodry" -> "hydrology"

---

## Referee Comment (RC2)

Review of "Numerical modeling of the tipping processes of ice detachment: a case study of Sedongpu Glacier in the Southeastern Tibetan Plateau"
By Zhang et al.
Submitted to *The Cryosphere*

Overview
Zhang and co-authors present a numerical model to represent glacier detachment, a rapid process that is not typically represented in physical glacier models – and a process that I believe many in the glaciology community are not likely familiar with. The novelty of this study is that it provides a framework for coupling the rapid evolution of internal stresses with basal friction, with great potential for applicability to improve hazard prediction in High Mountain Asia and other glacierized regions.

General comments
In general, the manuscript is well written and organized, with clear descriptions of the mathematical equations. The authors openly acknowledge the missing processes from this model that are also involved in glacier detachments, and justify their focus on internal stresses and basal sliding.

The paper includes links to a GitHub repository of model code and a Zenodo repository of data. The links are valid, but the repositories themselves need a bit of work to make them more usable. The Zenodo archive lacks a description of the files included, with no description of the variables saved in the MATLAB file or even a description of which simulation those results correspond to (out of several sensitivity simulations presented in the paper). In the GitHub repository, the README file does not provide any practical instructions for using the model scripts or descriptions of the different files included in the repository. I am also not sure if GitHub qualifies as a suitable repository for this journal or not.

Please see below for specific comments and questions that should be addressed to further strengthen this manuscript. Given the novelty of this topic and modeling approach, I recommend this paper for publication with some minor revisions. I enjoyed reading this paper, and I feel that it will make a great contribution to the glaciological literature, serving as a foundation for improved understanding and representation of dramatic detachment events in more complex glacier models.

Specific comments
Figure 1b,c – It would be helpful to draw on the broader map (Fig. 1a) the boundary of the DEMs to get properly oriented. You might consider including only one of the DEMs here, since they look so similar, but I can also understand the reasoning for including both. It's great that the caption points toward the difference DEM in a later figure.

Line 70 and Table 1: It would be helpful to provide a citation to justify the value chosen for A ($10^{-16}$), as it is several orders of magnitude higher than the values according to Cuffey/Paterson ($10^{-24}$).

Line 73: Is the basal friction coefficient held constant through time using the value from the inversion? (Section 3.1.1) Please clarify.

Figure 3b: The velocity looks very spotty, with alternating patterns of high and low velocity along the glacier center-line. This should be at least discussed, with some validation or justification for the pattern.

Figure 4: Are the values of basal friction prescribed to be 1000 at both ends of the glacier? What is the sensitivity to this value? Especially at the terminus, it seems like physically the friction should continue to be very low to allow for free movement.

Figure 5: It looks like there are numerical oscillations in the initial ice viscosity (5a) and also in the new steady state of mean speed and mean effective stress following the detachment event (5b). While they appear to be stable oscillations, it would be good to discuss these numerical artefacts. Did you experiment with using a smaller time step to resolve this oscillation?

Line 209: If ice thickness is being evolved, why does the cross-sectional profile look the same across all time steps in Figure 7?

Editorial comments
I am not going to identify individual typos and grammatical errors, but there are several throughout the paper.

Figure 6a: Where is the black line in this plot? Make it visible by using dashes or symbols if it is perfectly aligned with one of the other lines.

Figure 7: I suggest using the same colormap for all three cases for better comparison.

---

## Author Comment (AC1)

**Reply to reviewer 1**

GENERAL COMMENTS

The authors present a study of glacier detachment at Sedongpu Glacier from a numerical modeling perspective that attempts to recreate conditions of glacier attachment by simulating changes in glacier viscosity and feedbacks with basal sliding. The topic and motivation is really important, as glacier detachments deserve more scientific attention and can be destructive to downstream communities and infrastructure (as the authors note as well). I think the topic of the manuscript in itself has a lot of merit, and it's cool that the authors have been able to reproduce catastrophic collapse of the glacier. Because of this, I think that this paper is a good candidate for publication. However, I have serious concerns about the approach the authors take as well as the structure and presentation of the manuscript. These can be categorized in the following areas.

We thank the reviewer for the thorough review and also all of these great comments and suggestions! They significantly help in improving the manuscript!

1. I am somewhat unconvinced about the authors' choice to model glacier detachment from the perspective of a rate-weakening basal slip and variable ice viscosity. The authors claim that the feedback between ice stiffness and basal slip plays a major role in the instability that causes the glacier to detach within minutes in their simulations. However, there is very little discussion of how this feedback mechanism or its numerical implementation works. Neither have the authors cited any works that discuss this feedback. I remain unconvinced as to why this mechanism is responsible for the detachment both from a conceptual perspective and from their model results. The authors need to strengthen their explanation of this feedback and how/why they are modeling it. This could include adding a literature review section about damage mechanics and more explanation about why they decided to structure their coupled model the way they did.

Thanks for raising the questions. Here we present more details.
The glacier movement consists of two parts: basal sliding and internal deformation. For abrupt changes like glacier detachments, the basal sliding must experience a violent acceleration. According to Helanow et al. (2021), the universal basal sliding law with or without cavity probably has the same form as the Schoof sliding law:

$$\frac{\tau_b}{N} = C\left(\frac{u_b}{u_b + A_s C^n N^n}\right)^{1/n}$$

we can see that the basal sliding is affected by the basal effective pressure ($N$) that is related to basal water pressure ($P_w$),

$$N = \rho g H - P_w$$

where $\rho g H$ is the overburden ice pressure.

It means that as basal water pressure increases, the effective pressure will decrease and then helps accelerating ice flow. Thus, ideally, as $P_w$ approaches very closely to $\rho g H$, $N$ will be close to zero, and we imagine a very rapid increase in ice velocity. But things are a bit different than we thought. In the following two figures, we set $P_w = 0.01\rho g H$ and $P_w = 0.99\rho g H$, and we can clearly see the velocity increase.

[Figure]

Figure 1: The case of $P_w = 0.01\rho g H$

[Figure]

Figure 2: The case of $P_w = 0.99\rho g H$

However, it is still quite far away from the magnitude of ice detachment. We can get the reason by looking at the Schoof sliding law. For a given $\tau_b$, the basal velocity increases with a power of 1/n. Therefore, it is not very likely to have an abrupt ice detachment hazard by solely increasing basal water pressure in the Schoof sliding law.

After all, the Schoof sliding law is found under the traditional slow-moving dynamic framework (e.g., Gagliardini et al., 2007). As a result, the interaction between basal water pressure and slip is able to explain the mechanism of glacier surge (Thøgersen et al., 2019), but still hard to simulate ice detachment. In fact, if we take a look at some recent videos of ice detachment / avalanche, e.g., the recent Swiss glacier collapse (https://www.youtube.com/watch?v=KHXBVAnKDDY), we can see that glacier ice body no longer remain intact during the movement, i.e., it is highly damaged. Note that our Sedongpu case is a low-angle detachment, probably not as fractured as

the Swiss one. But it indicates clearly that the traditional constitutive law probably does not hold any more!

As we still lack a discrete element model for this problem, we hope to study the detachment events in the current continuum model framework. That is the reason we choose to use the constitutive law considering yield strength and plastic deformation in Bassis et al. (2021), which works for the damage mechanics for those highly crevassed regions in Antarctica ice shelves. This new constitutive law not only include the traditional elastic deformation, but also include the plastic deformation which occurs when stress exceeds yield strength of the material. This introduces a positive feedback - if stress exceeds yield strength, ice gets yielded, and a slow, elastic deformation becomes a fast, plastic deformation. In addition, the yield strength will also decrease as ice gets failure, leading to more easily yielded region. That will in turn get more ice yielded and eventually most of the glacier region becomes highly fractured.

Another important point is that, based on previous studies (Kääb et al., 2021), there are thick, soft till at the base of detached glaciers. For the Sedongpu Glacier in this study, the basal sediment is so soft that the subsequent erosion easily incised the bed up to several hundreds of meters. A soft bed helps accelerating ice flow and then make more ice regions get yielded, which will then result in more decreased basal frictions and accelerate more of ice flow.

In the revised manuscript, we add a new paragraph with more previous studies relevant to damage work and explanations.

"Previously, a well-studied fracture criteria that defines relationships between material strength and applied stresses has been widely applied in glaciology to model ice fracture and iceberg calving, as well as in studies of ice flow mechanics (Pralong and Funk, 2005; Albrecht and Levermann, 2012; Duddu and Waisman, 2012). Most numerical ice flow models adopt a stress threshold approach, where fracture occurs when stresses exceed a critical value (Hulbe et al., 2010; Borstad et al., 2016; Jiménez et al., 2017), though alternative methods like pressure or strain thresholds (Duddu et al., 2020) remain less utilized. Despite laboratory benchmarks, natural system observations to validate fracture criteria and stress thresholds remain scarce."

2. I am somewhat skeptical that the instability modeled in this manuscript (the viscosity-basal slip coupling) is truly a physical instability and not a numerical instability caused by the choice of model parameters. For example, when the coupling mechanism is activated, the viscosity of the ice goes from 60 to 0 Pa yr, which then triggers the rest of the glacier collapse. A more rigorous stability analysis is needed to ensure that this is a real physical phenomenon.

Thanks for the questioning. This is exactly what we've shown and discussed in Figure 3 (Figure 6 in the original manuscript). We agree that the modeled detachment results are dependent on the choices of model parameters, $\eta_{\min}$ (minimum viscosity) and $\tau_{\min}$ (minimum stress), in the following two equations.

$$\eta = \eta_{\min} + \left[\frac{1}{\eta_1} + \frac{1}{\eta_2} + \frac{1}{\eta_3}\right]^{-1}$$

$$\tau_y = \max\left(\tau_c - (\tau_c - \tau_{\min})\epsilon_p/\epsilon_c, \tau_{\min}\right)$$

Both $\eta_{\min}$ and $\tau_{\min}$ are for numerical stability, but their real values are hard to know.

[Figure]

Figure 3: The sensitivity of mean ice thickness changes of Sedongpu to different initial yield stress values (a), model parameters $\tau_{\min}$ and $\eta_{\min}$ (b) during the 25 minute model time span. The detachment begins at minute 5.

From Figure 3b we can see that the detachment speed dependents on the values of $\tau_{\min}$ and $\eta_{\min}$, but the choices of $\tau_{\min}$ and $\eta_{\min}$ do not change the fact of ice detachment occurrence - a result of plastic deformation due to ice strength decreases after reaching the yield strength. For example, if the yield strength of ice is pretty big, say, 1e7 Pa, then basically there will be no regions where ice stress exceeds yield strength and gets yielded. In this case, the glacier will stay stable (Figure 3a). But if the yield strength is small, say, 1e5 Pa, then during the ice movement, some regions on the glacier might be relatively easily to get yielded. After it becomes yielded, the yield strength will then further decrease and let it be even more vulnerable to get fractured. As a result, such regional, plastic, accelerating ice flow will quickly affect the whole glacier and lead to a detachment. We now add more discussions and explanations at Section " Model sensitivity".

3. The authors should provide some sort of validation or provide insights into the plausibility of their results. They repeatedly claim that the viscosity-basal slip feedback is more important than subglacial hydrology or till failure, but they do not provide any evidence to support this, either in the form of citing previous works or providing any sort of validation for their model. In addition, the authors do not show the misfit for their basal friction inversion, so we don't know whether or not their sliding parametrization is able to reproduce realistic glacier velocities.I think the authors should either include this evidence or scale back their claim that this feedback is more important than other mechanisms which have been more extensively proven to play a role in detachments

and surges (e.g., subglacial hydrology and geothermal heat).

We agree that model validation is an issue that needs more consideration. The ice detachment happened so quick that there was basically no time for in-situ observations. The satellite image data has been used in this study for reconstructing geometry and inversion of basal sliding parameters. The most reliable validation data is the duration time of ice detachment. According to "Scientific Assessment Report on the Ice Avalanche Dammed Lake Event at the Great Bend of the Yarlung Zangbo River" (in Chinese) published in 2022, the estimated duration time was around 300 seconds (Figure 4).

[Figure]

Figure 4: Seismic signal for 2018 Sedongpu detachment

In our model study, as shown in the black curve (Figure 3b), most of the ice volume is lost within around 5-6 minutes. As discussed in the previous bullet, $\tau_{\min}$ and $\eta_{\min}$ are not physical, but numerical parameters. However, these two parameters only affect the magnitude of ice detachment. Thus, we believe that our model mechanism is capable of simulating the abrupt changes of ice flow in this case. In the revised manuscript, we add some additional details about this issue.

Regarding the validation for inversion, we change the original Figure 4 to this one in below so that we can clearly see the performance of the inversion results.

[Figure]

Figure 4: The comparison of observed and modeled surface speed after inversion (a), and the inverted basal sliding parameter using the Robin inversion algorithm (b)

4. The authors don't include any sort of hydrology (subglacial or surface) in their modeling even though surface meltwater-driven hydrofracture and subglacial hydrology play a strong role in bulk

ice toughness and basal slip. I get that this would be difficult to include in a model like this, but it would add a lot to the manuscript if the authors could at least address this in the text. They also talk about effective stress but have not defined it any of their model equations, and claim that high effective stress corresponds to ice acceleration which goes against the traditional glaciology definition of effective stress.

Thanks for the suggestions. We agree that the discussions about subglacial hydrology is important. It is true that for now it is difficult to include basal hydrology module. We've discussed this issue in the section "Model limitations" (now changed to "Discussions"):
"Furthermore, our model does not incorporate thermal coupling or basal hydrology schemes, potentially neglecting key physical mechanisms involved in glacier detachment. For instance, Thøgersen et al. (2019) identified a velocity-strengthening-weakening transition that governs surge initiation, though their framework assumes intact ice and slow movement–conditions that may not adequately capture rapid detachment dynamics. While we acknowledge the importance of ice-bed interactions with basal hydrology (e.g., Schoof sliding law), our current implementation employs a simplified sliding law (Eqn 13) to couple basal till strength with ice flow. This approach, though computationally efficient, could be enhanced in future work to better represent these complex processes."
We also add the definition of effective stress at Section 3.1 "Ice flow model". The definition of effective stress is still traditional. During model runs, we calculate effective stress and use it to judge if it exceeds yield strength. If yes, then ice will go with a plastic deformation, which will then probably lead to a positive feedback of failure.

5. I would like a lot more information on the implementation of the model or finite element scheme. What mesh resolution is used, what meshing software was used, and how the spatial and temporal resolution was determined. In addition, there should be more discussion of the implicit assumptions that go into the model and what effects these have on the results.

The details of our model can be found in Wang et al. (2020) "A two-dimensional, higher-order, enthalpy-based thermomechanical ice flow model for mountain glaciers and its benchmark experiments". In the revised version of this manuscript, we add a bit more numerical descriptions, but we prefer not to repeat most of the numerical details here. The mesh resolution is clearly stated in Section 3.1. The choice of temporal resolution should at least satisfy the CFL condition. We use a very small dt (0.5 second) so that our model can remain stable in the forward runs. As we use a finite difference method, we do not use a specific meshing software as normally needed by a finite element method.

6. The manuscript doesn't follow as much of a logical flow as I would prefer. There is insufficient explanation of the model and its assumptions at the beginning of the manuscript before the equations are presented. I go more in detail about this in the specific comments below.

The structure of the paper is now adjusted according to the reviewer's suggestion.

7. I also believe it could be helpful for a native English speaker to read through the next iteration

of the manuscript as the language is often imprecise and not entirely scientific.

Language is improved in the revised manuscript.

SPECIFIC COMMENTS

L5: "Yield strength of the glacier" is very general. As the authors mentioned later in the manuscript, the yield strength of the glacier is extremely heterogeneous. I would recommend changing this to something like "initial yield strength of glacier ice".

Thanks. Changed.

L8: "just several model time steps" is pretty vague

Changed to "The transition from slow to abrupt flow occurs after most regions of the glacier reach a plastic state.".

L9-10: How would these results be used for early warnings? Maybe you can talk about this later in the discussion.

This sentence is removed and some discussions are added in the "Conclusion" section.

L16: Merge this with the previous paragraph

Merged.

L22: The "largest ever recorded event" of what? Of surges, of a glacier detachment? The language could be a bit more precise.

The context is now updated.

L24: The authors mention that remote sensing advances allow us to detect changes in surface features like crevasses and fractures but the authors did not use any of these data to validate the model results

Right. Thanks for pointing this out. The detachment occurred so quick that we basically do not have any remote sensing data for validation. As we replied earlier, we used the duration of detachment for validation. We now remove this paragraph.

L38: This is the first time the authors mention the ice stiffness-basal slip positive feedback. It would be great to have an additional few paragraphs explaining this feedback and motivating why the authors choose to model it. It would also be great to see some previous examples where this mechanism was suggested or modeled successfully before.

We add a new paragraph in the revised manuscript:

Glacier fracture and damage significantly accelerate ice flow by structurally weakening ice and reducing its effective bulk viscosity, as observed in Pine Island and Thwaites Glaciers where upstream fracturing correlates with flow acceleration (Lhermitte et al., 2020; Sun and Gudmundsson, 2023; Surawy-Stepney et al., 2023). This damage interacts with basal slip—where ice slides over bedrock—through stress redistribution that enhances basal crevassing (Bassis and Ma, 2015) and by facilitating meltwater penetration, which reduces basal friction and further accelerates slip (Sun et al., 2021; Clayton et al., 2022). Consequently, damage evolution is critical for projecting long-term ice flow changes and land ice stability (Albrecht and Levermann, 2014; Bassis et al., 2024), though model uncertainties persist regarding damage parameters and feedback mechanisms.

L41: Specify the glacier name again

Specified.

L44: What is Medog? Is it a village, a weather station?

It is the name of a county. This information is added.

L48: Some information about the quality/texture of the Quaternary deposits would be useful here. Is it soft sediment or more coarse till? Also, how thick is the debris cover on the glacier?

We add a new sentence here: The Sedongpu Glacier was underlain by a thick sediment/moraine layer which was eroded during the 2018 detachment event, forming a canyon up to 300 meters deep (Kaab, 2021; Kaab and Girod, 2023).

Figure 1: (a) It would be great to highlight the glacier that is being modeled. (b) and (c) the text is a little blurry - would be better to have higher resolution. Please also show the outline of the glacier in (b) and (c) so we can get a better idea of where the glacier is inside the valley. Plotting the difference between the two DEMs would also probably be more useful (i.e., 2018 DEM minus 2015 DEM).

Thank you for your suggestions. We have emphasized the location of Sedongpu in figure a, upgraded the resolution of figures b and c, and also added glacier contour information to these two figures. In addition, we have added the elevation difference (d) for 2015-2018.

L56: I think this whole section (ice flow model) should go after Figure 2.

Thanks for the suggestion, but we prefer the current way. This figure is a overall workflow, so it might be a bit easier for people to get the idea after we introduce the model.

L57: What is the dimensionality of this model? It would be good to specify.

We stated very clearly that it is a two dimensional model.

L57: PoLIM is a thermomechanical model. What are the thermal parameters specified here? What assumptions are made about heat flow? Is there any geothermal heat flux?

Right, PoLIM is a thermomechanical model, which means the velocity and temperature solver could be coupled together. We can also choose not to solve the temperature field by assuming a constant flow rate parameter (A) for temperate glaciers. Sedongpu Glacier is a typical temperate glacier. So we just need to use a constant A. In this case, we do not have to prescribe geothermal heat flux, which is a boundary constraint for temperature solver.

L70: Reference to Table 1 would be useful here.

Add the reference to Table 1.

L71: Justify why you are using a Weertman-type sliding law here, and why effective pressure (N) isn't important.

We use a linear Weertman sliding law prior to detachment. The main reasons are (i) we do not have any data of basal water pressure and (iii) we just need an initialization that captures the ice flow features prior to detachment so that we can well simulate the subsequent dynamics, which is based on a good model initialization.

L83: There is also another model initialization section in the "results and discussion" (Section 5.1). Consider merging these two?

We merged the subsection 5.1 "Model initialization" into 5.2 "Sedongpu detachment simulation" as initialization is also part of our simulations so that we can avoid this confusion and also.

Table 1: What is the geothermal heat flux? Also, consider converting the rate factor A to Pa^-n s^-1 for easier comparison with other papers. It would also be good to include citations for the choices of critical strain and intact strength of the ice.

The unit of A is changed. We do not really need geothermal heat flux in this case, as Sedongpu is a typical temperate glacier and we can use a constant A, i.e., we do not really need to solve the temperature field. As a result, we do not need to use geothermal heat flux as a boundary thermal condition. The citation for critical and intact strength is also added.

L90: Where do you get the Dirichlet and Neumann velocities from? What does it mean for velocity fields to be Neumann/Dirichlet? Also, what is the misfit from this inversion, and is the model able to reproduce realistic velocities?

Great you ask this! To fully understand these questions, I have to refer the Arthern and

Gudmunsson (2010) paper. Here are some brief and simplified explanations:

The basic idea of the Robin inversion is to compare the observed and modeled ice surface velocity by changing basal frictions each step. For example, at some location marked as $x_i$, we can first set an initial basal friction $\beta$, and then we run the model. If we see the modeled surface velocity is larger than the observed value. Then $\beta$ is probably too small, and we need to adjust its value by increasing it a bit, and then we run the model again. There are chances that this time we get a smaller surface velocity than observed, and then we need to reduce $\beta$ a bit. We will repeat such iterations until we have modeled velocities that are closed to observations. So now we understand that there are two different surface velocity values involved here, modeled and observed surface velocity. For Dirichlet boundary condition, we apply observational velocities from remote sensing data and apply them as surface boundary condition, $u_s = u_{obs}$. For Neumann boundary condition, we apply a normal stress free condition at ice surface, just as a normal ice flow model does, $\sigma \cdot n = 0$. So we know that our model can reproduce realistic velocities if modeled velocities match observations after initialization.

L101: There needs to be an explanation of how you obtain eta_1 and eta_2 and what each of the three viscosities mean. Also, how is eta_min determined and what is the model's sensitivity to choice of eta_min?

We now change the description way of this equation. $\eta_{min}$ is a tuning parameter and its impact on model results is shown in Figure 6b. Generally speaking, $\eta_{min}$ is the minimum viscosity, so the modeled ice velocity will increase if we set a small $\eta_{min}$.

L106: When and how does the viscosity transition to plastic viscosity?

The ice will become "plastic" if / when the stress exceeds yield strength. See this diagram.

[Figure]

Source: https://byjus.com/physics/yield-strength/

L108: It would be helpful to cite previous works that have demonstrated good results with a model like this

Citation added.

L113: Are you using the linear Weertman-type or this rate limited one?

We use the linear Weertman sliding law just for model initialization. To simulate ice detachment, we use this sliding law instead in order to consider the impact of stress transitioning.

L117: Include a reference to Figure 2 before presenting the model equations. It would also be great to expand more on Figure 2 in the text.

Manuscript is changed here according to these suggestions.

Equation 13: Why is the ice yield strength in the expression for basal shear stress? In Bassis et al.'s supplement, the second term inside the brackets has tau_c (the intact strength) in the denominator, but here you have tau_y (yield strength).

Right. For the equation in Bassis et al. (2021), the bed is probably not as soft as in our Sedongpu case. See the following figure. The bed of Sedongpu Glacier is deposited by up to 300 m soft tills. Thus, we here use \tau_y to better represent the coupling of ice stiffness and basal slip. We now put some more sentences here in the manuscript.

[Figure]

L126: The "Datasets" section should be before the model equations.

Changed.

L138: Need more information on what the "glaciological modeling results" are. Are they using a

flow approximation? If so, what is being used? Some more details on how the glacier geometry was determined would be helpful here.

Thank you for this helpful comment. In the revised manuscript, we have clarified the method used to obtain the distributed ice thickness of Sedongpu Glacier. Specifically, we used the GlaTE software (Langhammer et al., 2019), which derives glacier-wide ice thickness by optimally combining discrete thickness estimates with glaciological modeling constraints through an inversion framework. The modeling component is based on the method of Clarke et al. (2013), which uses a shallow-ice approximation to relate basal shear stress to surface slope and apparent mass balance. This formulation is cast as a linear optimization problem with smoothness regularization, resulting in a physically plausible thickness field.

In our case, the discrete thickness estimates were derived from elevation differences between pre-detachment glacier surfaces and post-detachment exposed bed topography, providing first-order constraints on ice thickness. These thickness points, together with a DEM and glacier outline, were used as input for GlaTE. The resulting distributed ice thickness field was then sampled along a centerline generated using the method of Kienholz et al. (2014), and this flowline geometry was subsequently used in the PoLIM simulations. These clarifications have been incorporated into the revised manuscript.

L139: What are some limitations to using a flowline model as you are doing, compared to a 2D model? How would this change the results?

I guess you'd like to know the comparison with a 3D model. Our flowline model is 2D in x-z dimensions. I would say that using a 3D model could be more exciting, since it uses a 3D geometry and possibly more accurate than a 2D flowline model. But it will also bring more difficulties in numerical implementation and modeling, for example, meshing (finite element) and accurate data inputs. The major limitation of using a 2D model is that it does not fully account for the impact of lateral drag of ice flow. But the main dynamic features should be similar between a 2D and 3D model, if we implement the same instability physics in the model.

L144: Why were higher velocities filtered out? It seems like a threshold of 400 cm/day could exclude important data if the glacier was moving faster.

At the initial velocity calculation completion, some areas of extreme values (greater than 400 cm/day) were identified: these results were generally concentrated in the upper part of the glacier (cloud cover); there were also some anomalies in the lower and middle part of the glacier with a very small area range, which were found to be practically not undergoing rapid displacements by comparing the two-phase optical images of this location. Therefore, we consider results greater than 400 cm/day as anomalies.

Figure 3: Move the panel labels (a and b) to above the figures so they are more visible. Including a hillshade contour around the glacier domain would also be useful. In Panel A, you should use a diverging colormap so it's easier to see where the glacier thickness increased and decreased. Also,

why is there data missing in the upper tributary in panel B?

We have moved the panel labels (a and b) to above the figures and added the hillshade layer in this figure. Due to cloud interference in the upper glacier region, we unable to calculate the surface velocity of this region, resulting in no valid output for this region.

L148: There was already a model initialization section. This should be merged with that. It doesn't make sense to have a model initialization in the Results/Discussion section.

This section is now merged into its following model results section.

L153: Why did you use the mean velocity from 2015-2018 instead of the velocities just prior to the detachment? Does Sedongpu exhibit seasonal variations, and how would that affect your calculation of the friction coefficient at the moment of detachment?

It is a nice question. The model initialization mainly account for an average status of land ice dynamics. Please see Seroussi et al. (2020) "initMIP-Antarctica: an ice sheet model initialization experiment of ISMIP6" for more details. The inputs (geometry, velocity) for model initialization are data obtained across multi-year time spans. As we explained the Robin inversion at L90, the model initialization is basically a numerical optimization approach that could tune basal friction parameters to match observations. It is hard for an inversion algorithm to capture fast and transient processes. Glacier movements all have seasonal variations, but the remote sensing data can only get the mean velocity value for a given period of time. Thus, it is also pretty hard to accurately simulate the high-res seasonal changes before the detachment. This means the stress regime across the glacier before the detachment probably also represents an average condition. It might impact the threshold of ice stress that exceeds yield strength. But in this study, we think the mechanism explanation is a bit more important than model accuracy.

Figure 4: What is the resolution of this flow line model? How do these velocities compare to observations?

Please see our descriptions in the model description section: We use a $\Delta x$ = 48 m in x and 20 vertical layers in z with a terrain-following coordinate. Now we change Figure 4 so that we can see a direct comparison with surface velocity.

L159: The authors claim that external environmental forcings could be responsible for the detachment, but these forcings are not included in the model.

Right. We now change this sentence to "Environmental forcings may have acted as external triggers for the 2018 Sedongpu detachment". So it is clear that they might induce the occurrence of detachment, but they are not the responsible mechanism.

L164: Why is the coupling mechanism activated at t=5 minutes? What physical event does this represent? Does the model reach any sort of equilibrium prior to t_0? And why does the glacier

instantaneously collapse when you activate the coupling mechanism - can you be sure that it isn't a numerical instability?

Before we simulate ice detachment, we need to confirm our model is stable. We explain this a bit more in the manuscript. But 5 min is an arbitrary choice. It should be long enough for the model to get stabilized before the detachment. The instantaneous collapse after we activate the instability mechanism is exactly what we'd like to see and is the goal of this paper. The real Sedongpu glacier detachment just happened this way. We explained in the major comment responses why it is a physical result, not numerical instability.

L167: 90,000 m/hour seems unrealistic. Can you comment on why it's so high, and what this could be representing?

Unfortunately, it is real. The whole detachment finished in 300 seconds. In a warming world, some glaciers may not move slowly any more. There truly exists some tipping points that those glaciers may just disappear in a sudden. I am not sure so far if we will have similar events in Antarctica, but these detachments are real warnings for the whole human society.

L170: This is the first mention of effective stress in the entire paper. There is no mathematical or conceptual definition of effective pressure anywhere prior to this, and effective stress was not a parameter in any of the model equations shown. How is effective stress being calculated?

The definition of effective stress is now added in the model description section.

L171: An increase in effective stress should not result in ice acceleration, unless you are defining effective stress differently from the traditional glaciological definition. Higher effective stresses are associated with higher frictional contact between the ice and bed, and so a higher effective stress should correspond with the glacier slowing down. Effective stress needs to be defined explicitly in the ice flow model section and maybe again here.

You might understand it in a reversed way. The increase of effective stress is a result of ice acceleration. If we look at the equation of stress, $\tau = \eta \dot{\epsilon}$, stress will increase if strain rate increases when ice flow speeds up if viscosity keeps stable. But the viscosity decreases in our cases. The changes of stress is a combination result of the changes of viscosity and strain rate. Clearly, in our case, the increase of strain rate overwhelms the decrease of viscosity, and after the increased stress exceeds the yield strength if ice, the detachment instability mechanism is triggered.

L173: Is the 6.3 minutes from the beginning of the simulation or 6.3 minutes after t_0 the activation of the coupling mechanism?

It is 6.3 mins after t_0 when we activate the coupling mechanism. We now add "after t_0" at the end of this sentence.

Figure 5: It would be great if the authors could comment on the sudden change in viscosity from 60 to 0 Pa yr. It seems unrealistic to me that the viscosity would go to 0 within less than a minute.

The viscosity does not reduce to 0, but to $\eta_{min}$. Perhaps you could take a look at this recent Swiss glacier collapse (https://www.youtube.com/watch?v=KHXBVAnKDDY). The glacier became highly fractured during the high-speed movement. In such cases, ice viscosity can no longer remain a big value. We can also look at Figure 6b. If we increase $\eta_{min}$, the glacier collapse will also take a longer time. I agree that this kind of events are a bit beyond our imagination. They are quite different than traditional slow-moving glacier dynamics.

L177: Nice relating with previous work

Thanks. There are very few previous studies about glacier detachment, unfortunately.

L184: If yield strength varies from 100-1000 kPa, why did you only test between 300-500 kPa?

Right. This sentence is not accurate. We change it to "Figure 6a shows cases with initial yield strengths between 300 and 500 kPa". We've tested more cases than we showed here. But as the glacier detachment occurs at an initial yield strength of around 430 KPa, it is not very necessary to show other cases like 100 KPa and 200 KPa, etc…

L190: What is the sensitivity of the model to A and to the temperature of the ice?

The A value (3.17e-24 Pa^{-3} s^{-1}) we use in this study is for ice temperature around 0 degree C. See the following table from Cuffey and Pateson (2010) "The Physics of Glaciers". We can get some slower collapses if we apply smaller A values, but I am not sure if it is necessary to do that.

Table 3.3: Measured and inferred values of creep parameter $A$ at different temperatures, for $n = 3$.

| $T$ (°C) | $A$ ($10^{-25}$ s$^{-1}$ Pa$^{-3}$) | Method | Reference |
|---|---|---|---|
| 0 | 24 | Mean of 5 calibrated models | See below[†] |
|  | 38 | Mean of 5 borehole tilt values° | Raymond 1980 |
|  | 55 | Closure of tunnels | Nye 1953 |
| 0 | 24 | **Recommended base value** |  |
|  | 93 | Various lab tests | Budd and Jacka 1989 |
| –2 | 27 | Various lab tests | Budd and Jacka 1989 |
| –10 | 3.9 | Ice shelf spreading | Jezek et al. 1985 |
|  | 5.3 | Ice shelf spreading | Thomas 1973b |
|  | 2.5–4.3 | Ross Ice Shelf flow[‡] | MacAyeal et al. 1996 |
|  | 1.8–3.2 | Filchner-Ronne Ice Shelf flow[‡] | MacAyeal et al. 1998 |
| –10 | 7.6 | Borehole tilting | Fisher and Koerner 1986 |
|  | 6.7 | Flow-line with borehole | Reeh and Paterson 1988 |
|  | 8.7 | Borehole tilting | Dahl-Jensen and Gundestrup 1987 |
| –10 | 3.8 | Mean of ice shelf values |  |
|  | 7.7 | Mean of simple shear values |  |
| –10 | 3.5 | Various lab tests | Budd and Jacka 1989 |
|  | 3.5 | **Recommended base value** |  |

[†] Hubbard et al. 1998; Gudmundsson 1999; Adalgeirsdottir et al. 2000; Albrecht et al. 2000; Truffer et al. 2001.
° We have calculated $A$ for $n = 3$ using $AR^{0} = A_o \tau_o^{n}$, where $A_o$, $\tau_o$, and $n$ are values given by original authors, and $R$ is Raymond's corrected stress value. Stresses are for the greatest depth of reported measurements.
[‡] Calibrated model for flow of entire ice shelf (Ross) or part of ice shelf (Filchner-Ronne). Low and high values are for effective temperatures of −15 and −20 °C, respectively.

L199: It would be great to include a conceptual explanation or discussion of why the coupling

feedback mechanism almost instantaneously results in failure and rapid acceleration of the ice. This is the crux of the manuscript and should be explained in more detail.

The section "The tipping processes of Sedongpu detachment" presents details of this coupling feedback. We now put more information to make it more clear.

L200: Earlier the authors suggested that hydrology doesn't play as much of a role as the viscosity-slip feedback mechanism. It is unclear to me then why rainfall events should be monitored as opposed to crevassing or softening of the glacier ice.

Here we mean external factors like heavy rainfall could be incentives for glacier detachment to happen. It could lubricate ice bed, accelerate ice flow and soften glacier ice. There are chances that, during these processes, the ice stress at some locations might exceeds yield strength and then turn on the instability mechanism. But yes, we believe in this case water (hydrology) is not as important as the yield-slip coupling mechanism.

Figure 6: What is the normalized value? I don't think the normalization was defined clearly in the text or in the caption for the figures. Is it a spatial average? Maybe I missed it, but I don't understand the y-axis for these plots.

We add an additional sentence in the caption: "The y-axis indicates the normalized value of mean ice thickness along the flowline." We use normalized value here in order to better see how much ice mass is lost during the model run.

L206: What does ice becoming "yielded" mean?

It means the stress goes beyond the yield strength of ice. But we change it to some other expressions like "exceeds yield strength" to avoid potential confusion.

L218-220: This sentence would make more sense if it were in the introduction. Or, if the authors wish to claim that the elastic-plastic transition is important, it would be helpful to include a scatter plot of stress vs. strain rate or something like that to show the glacier viscosity transitioning from elastic to plastic.

Now we add a new figure showing the relationship between basal drag, effective pressure and basal geometry (Iken's bound). Also see the reply to L237 below.

L224-225: "This has remarkable scientific and engineering applications for large infrastructures in the local regions" - I think it would be good to be more specific about which aspects of your results would be helpful for science/engineering applications. Otherwise, this sentence may make more sense if it were in the introduction or conclusion.

Nice point (reminder). We now remove this sentence and replace it with another one "To do that, we need to conduct detailed and accurate investigation of glacier geometry for building a reliable

ice flow model and early warning system". This is a sensitive region for big infrastructures, and we better just discuss science here.

Figure 7: The panel labels are hard to see - maybe put them on top of each plot instead of at the bottom. Also the colorbars need units.

Figure 7 is now improved and the caption is also changed accordingly.

L226: This section should go closer to the beginning when you are introducing the model - it would be helpful to understand more of the model limitations from the outset.

Thanks for the suggestion. We prefer to let this section stay here, as we feel it is more like a supplement to both ice flow model and model results.

L228: The assumption of constant ice density wasn't included in the model description at the beginning. It would be helpful to mention this in the "ice flow model" section.

Added.

L237: Could you write briefly about how including some of these mechanisms could affect the results? E.g., if you used Schoof's sliding law or Iken's bound, how would that change the results?

We now add a new paragraph in the "Model limitations" (now changed to "Discussions") section regarding the Iken's bound:
In kaab et al. (2021), they analyzed the force balance of simplified, slab geometries and marked Sedongpu Glacier as stable. In fact, for stable glaciers with basal cavities, basal drag is constrained by an upper limit known as Iken's bound,
$$\tau_b/N \le m_{\max}$$
where $\tau_b$ is the basal shear stress, N is the effective pressure and $m_{\max}$ is the maximum value of the up-glacier-facing slopes of obstacles. Here we assume $N$ is the overburden ice pressure (no basal water pressure) and set $m_{\max}$ to the maximum bed slope. Figure 8 shows that once the detachment instability mechanism is triggered at t_0, the ratio $\tau_b/(Nm_{\max})$
 in the upstream region of Sedongpu Glacier rapidly exceeds 1 (violating Iken's bound), which aligns closely well with the timing of the detachment event.

The advantage of the Schoof law is that we can include the basal hydrology. In the Sedongpu case, it is hard to do as we do not consider the effect of basal water, which is clearly stated in the context.

L248-249: What are the early warning signals of glacier detachment? These weren't discussed at all in the manuscript.

We revised the "Conclusion" section a bit and put some more explanations of the earning warning signals.

Technical corrections and typos

L38: "novel" -> "novel", "postive" -> "positive"

Corrected.

L101: "Glen's pow law" -> "Glen's power law"

Corrected.

L89: change "According" to "Following"

Corrected.

L163: "simulated" -> "simulate"

Corrected.

L183: "destablizing" -> "destabilizing"

Corrected.

L230: "descrete" -> "discrete"

Corrected.

L234: "hydrolodry" -> "hydrology"

Corrected.

---

## Author Comment (AC2)

**Reply to reviewer 2**

**Overview**

Zhang and co-authors present a numerical model to represent glacier detachment, a rapid process that is not typically represented in physical glacier models – and a process that I believe many in the glaciology community are not likely familiar with. The novelty of this study is that it provides a framework for coupling the rapid evolution of internal stresses with basal friction, with great potential for applicability to improve hazard prediction in High Mountain Asia and other glacierized regions.

We thank the reviewer for the thorough review and also all of these great comments and suggestions! They significantly help in improving the manuscript!

**General comments**

In general, the manuscript is well written and organized, with clear descriptions of the mathematical equations. The authors openly acknowledge the missing processes from this model that are also involved in glacier detachments, and justify their focus on internal stresses and basal sliding.

We thank the reviewer for this positive comment.

The paper includes links to a GitHub repository of model code and a Zenodo repository of data. The links are valid, but the repositories themselves need a bit of work to make them more usable. The Zenodo archive lacks a description of the files included, with no description of the variables saved in the MATLAB file or even a description of which simulation those results correspond to (out of several sensitivity simulations presented in the paper). In the GitHub repository, the README file does not provide any practical instructions for using the model scripts or descriptions of the different files included in the repository. I am also not sure if GitHub qualifies as a suitable repository for this journal or not.

We now modify the code and data availability sections. We put a version of the code for simulating Sedongpu Glacier detachment at https://zenodo.org/records/15831881. By this we hope people can freely try running the code. In the code folder, we put a readme document for a guide of using PoLIM. At the same location, we put velocity and glacier boundary data for Sedongpu Glacier.

Please see below for specific comments and questions that should be addressed to further strengthen this manuscript. Given the novelty of this topic and modeling approach, I recommend this paper for publication with some minor revisions. I enjoyed reading this paper, and I feel that it will make a great contribution to the glaciological literature, serving as a foundation for improved understanding and representation of dramatic detachment events in more complex glacier models.

We thank the reviewer for this positive comment and endorsement!

**Specific comments**

Figure 1b,c – It would be helpful to draw on the broader map (Fig. 1a) the boundary of the DEMs to get properly oriented. You might consider including only one of the DEMs here, since they look so similar, but I can also understand the reasoning for including both. It's great that the caption points toward the difference DEM in a later figure.

We have indicated the approximate range of the DEMs (a) and also included the elevation difference (d) from 2015 to 2018 in Figure 1.

Line 70 and Table 1: It would be helpful to provide a citation to justify the value chosen for A (10^-16), as it is several orders of magnitude higher than the values according to Cuffey/Paterson (10^-24).

The $10^{-16}$ value is with the unit of $Pa^{-3}$ $yr^{-1}$. We now change it to the unit of $Pa^{-3}$ $s^{-1}$ (3.17e-24). This value is for ice temperature around 0 degree C according to Cuffey and Paterson (2010). See the table below. The citation is added.

**Table 3.3: Measured and inferred values of creep parameter $A$ at different temperatures, for $n = 3$.**

| $T$ ($°C$) | $A$ ($10^{-25}$ $s^{-1}$ $Pa^{-3}$) | Method | Reference |
|---|---|---|---|
| 0 | 24 | Mean of 5 calibrated models | See below[†] |
|  | 38 | Mean of 5 borehole tilt values[°] | Raymond 1980 |
|  | 55 | Closure of tunnels | Nye 1953 |
| 0 | 24 | **Recommended base value** |  |
|  | 93 | Various lab tests | Budd and Jacka 1989 |
| –2 | 27 | Various lab tests | Budd and Jacka 1989 |
| –10 | 3.9 | Ice shelf spreading | Jezek et al. 1985 |
|  | 5.3 | Ice shelf spreading | Thomas 1973b |
|  | 2.5–4.3 | Ross Ice Shelf flow[‡] | MacAyeal et al. 1996 |
|  | 1.8–3.2 | Filchner-Ronne Ice Shelf flow[‡] | MacAyeal et al. 1998 |
| –10 | 7.6 | Borehole tilting | Fisher and Koerner 1986 |
|  | 6.7 | Flow-line with borehole | Reeh and Paterson 1988 |
|  | 8.7 | Borehole tilting | Dahl-Jensen and Gundestrup 1987 |
| –10 | 3.8 | Mean of ice shelf values |  |
|  | 7.7 | Mean of simple shear values |  |
| –10 | 3.5 | Various lab tests | Budd and Jacka 1989 |
|  | 3.5 | **Recommended base value** |  |

[†] Hubbard et al. 1998; Gudmundsson 1999; Adalgeirsdottir et al. 2000; Albrecht et al. 2000; Truffer et al. 2001.
[°] We have calculated $A$ for $n = 3$ using $A R^8 = A_o \tau_o^n$, where $A_o$, $\tau_o$, and $n$ are values given by original authors, and $R$ is Raymond's corrected stress value. Stresses are for the greatest depth of reported measurements.
[‡] Calibrated model for flow of entire ice shelf (Ross) or part of ice shelf (Filchner-Ronne). Low and high values are for effective temperatures of −15 and −20 °C, respectively.

Line 73: Is the basal friction coefficient held constant through time using the value from the inversion? (Section 3.1.1) Please clarify.

Right. We set \beta constant in space before we simulate the detachment. We now add an additional sentence to make it more clear.

Figure 3b: The velocity looks very spotty, with alternating patterns of high and low velocity along the glacier center-line. This should be at least discussed, with some validation or justification for the pattern.

ImGRAFT's methodology primarily relies on feature point tracking for surface velocity estimation. However, due to drastic variations in surface morphology, posing substantial challenges for feature point tracking in this region. Consequently, the estimated velocity results exhibit considerable errors in certain areas and demonstrate pronounced spatial heterogeneity. Nevertheless, the current findings generally capture the overall surface velocity distribution pattern of this glacier.

Figure 4: Are the values of basal friction prescribed to be 1000 at both ends of the glacier? What is the sensitivity to this value? Especially at the terminus, it seems like physically the friction should continue to be very low to allow for free movement.

Very good point. We do prescribe an initial $\beta$ = 1e3 Pa m$^{-1}$ yr. At the glacier head and terminus, we use a Dirichlet boundary condition in the velocity solver, so $\beta$ will not be updated during the initialization. Here I present three sensitivity plots for initial $\beta$ = 1e2, 1e3 and 1e4 Pa m$^{-1}$ yr. We can see clearly that the initial value has some impacts on grids close to the head and terminus, but the majority of glacier is not affected. To avoid confusion, we do not plot $\beta$ for the head and terminus grids in the revised manuscript.

[Figure]

Figure 1: initial $\beta$ = 100 Pa m$^{-1}$ yr

[Figure]

Figure 2: initial $\beta$ = 1000 Pa m$^{-1}$ yr

[Figure]

Figure 3: initial $\beta$ = 10000 Pa m$^{-1}$ yr

Figure 5: It looks like there are numerical oscillations in the initial ice viscosity (5a) and also in the new steady state of mean speed and mean effective stress following the detachment event (5b). While they appear to be stable oscillations, it would be good to discuss these numerical artefacts. Did you experiment with using a smaller time step to resolve this oscillation?

Thanks for this nice suggestion. We did several tests with smaller time steps (0.2s, 0.5s), and indeed, the oscillation can be greatly reduced by using a small time step. After some digging, it is probably due to some numerical instabilities raised by the basal sliding boundary condition we implemented in the model. We now use the figure below with a time step = 0.5 s for the consideration of computing time.

[Figure]

Line 209: If ice thickness is being evolved, why does the cross-sectional profile look the same across all time steps in Figure 7?

Right, those are forward runs and ice thickness is evolving. It is just because the time step is so small (1 second) that we can just barely see the changes within three time steps.

**Editorial comments**
I am not going to identify individual typos and grammatical errors, but there are several throughout the paper.

Thanks. We now carefully check the manuscript and correct all of the typos.

Figure 6a: Where is the black line in this plot? Make it visible by using dashes or symbols if it is perfectly aligned with one of the other lines.

Thanks for the suggestion. We now change the red solid curve to dashed in Figure 6 so that the black curve show up clearly.

Figure 7: I suggest using the same colormap for all three cases for better comparison.

Figure 7 is updated.

---

## Referee Report (RR1)

**"Numerical Modeling of Ice Detachment Tipping Processes: Insights from the Sedongpu Glacier, Southeastern Tibetan Plateau" by T. Zhang et al. (2025)**

Major comments

"Numerical Modeling of Ice Detachment Tipping Processe: Insights from the Sedongpu Glacier, Southeastern Tibetan Plateau" by Zhang et al. (2025) uses a numerical modeling framework to demonstrate an ice weakening-basal slip feedback that results in the rapid detachment of Sedongpu Glacier which was observed in 2018.

To the authors, thank you very much for responding to all my comments in the last review. The explanation of the science you did is so much clearer now and the manuscript is much better organized. The sections flow much more effectively together. Now, readers can much better appreciate the novelty and excitement of this paper! I think the paper is well on its way to being accepted for publication, but I'd like to offer some more minor revisions.

One remaining thing I would like the authors to clarify is the following. There have been some references to it in the updated manuscript, but I think it should be fleshed out a little bit more: What exactly is it that triggers the dramatic ice weakening mechanism mentioned in L213? Obviously, in real life, you cannot simply "switch on" a new piece of physics (i.e., the ice weakening-basal slip feedback) – so what natural event(s) could cause this mechanism to be activated? You have made it clear that activating this mechanism with a high initial yield strength will not trigger glacier detachment, but reasons the mechanism itself is activated are not as clear. I would love to see this a little more fleshed out in the Discussion section. It would also be nice to include a little discussion on why glacier detachment occurred at Sedongpu Valley at this particular area, yet is relatively uncommon (so far) in other glaciers of High Mountain Asia. If you don't know, it would be great to identify this as an important knowledge gap.

Additionally, in the first set of reviews I misunderstood the authors' definition of "effective stress" because it was not defined in the text. It was not until they actually provided their definition in the second draft that I understood where this misunderstanding came from. Many glaciologists will first associate the phrase "effective stress" or "effective pressure" to be N, the overburden pressure minus water pressure (rho g h – p_w), which is directly related to ice speeds in sliding laws such as Budd's and Schoof's, although what you are referring to as "effective stress" is technically the correct term. I think you may want to add a sentence clarifying this by emphasizing the relationship between effective stress and deviatoric stress. Ideally, I would consider referring to deviatoric stress instead of effective stress in your discussions, as this will avoid confusing readers for whom the concept of deviatoric stress is more intuitive. Alternatively, please include a small note such as "(Note:

effective pressure here is defined from effective strain rate and is not related to the effective pressure N)".

Specific/minor comments

A small request: When adding new information (citations, clarification, new phrasings, etc.) in response to comments, please copy those changes into the response document and write what line they are at in the updated manuscript.

L4: "incorporating a positive feedback mechanism between ice stiffness and basal slip"

L40: I think you can probably just cite Bassis et al. (2021)

L42: "understandings" -> "understanding"

The introduction generally flowed so much better than it did in the first draft. I am able to understand your motivation for the science much better than before. Your explanation of glacier detachment is also much more effective than before.

L45: You can probably take out the last sentence

L63: Delete "Glacier topography:"

L73: ensure that your in-text citation is formatted correctly with parentheses

L77: What kind of regularization does the program use? If just a simple Tikhonov, you could just say "...a linear optimization problem with a Tikhonov smoothness regularization to produce..."

L80: what are the errors associated with the cross-correlation method for generating surface displacements? This could be mentioned here or in the section where you talk about other model limitations.

L88: Thank you for providing more details about the model! These are very useful.

L90: I asked in the last review about the thermal constraints on the model. Please explain in the text that you neglected thermal evolution and kept A constant, and that there is no geothermal flux (as you wrote in your responses).

Figure 1d: It would be great to overlay the outline of the glacier on this panel so we can contextualize these surface elevation changes

L102: Assuming the definition of effective strain rate in Cuffey & Paterson Eq. 3.17, your definition of effective strain rate does not make sense. If your model is 2D, it doesn't make sense that you'd have strains in 3 directions. Also, not sure where the third term (cross

term) on the right-hand side comes from. Could you please clarify why you have defined effective strain rate this way?

L118: Till deformation is also part of this equation – some glaciers don't slide along their beds or deform much internally, and their motion is solely due to the plastic deformation of till underneath. If you're considering till deformation to be part of basal sliding, you may want to specify this more explicitly rather than leaving it out!

How do you know that some of observed detachment is not just due to very fast till deformation? You said that glacier motion has two parts – basal sliding and internal deformation. But till deformation should be included in this. You could clarify that you are including till deformation lumped in with the friction coefficient, but you should then also address that this friction coefficient will not stay constant with time – how is this addressed?

L126: In the first review, I asked about Dirichlet and Neumann boundary conditions not because I didn't know what they are, but because the text at this place is misleading. Arthern et al. (2015) solves the momentum equations by using both the stress-free boundary conditions (Neumann) and observed velocities (Dirichlet). Here, you have said that Dirichlet and Neumann boundary conditions are observed and modeled ice velocities, which is a little confusing, since surface velocity is a Dirichlet condition. I would suggest reframing this closer to the way Arthern et al. (2015) defines it.

Table 1: Could you please clarify why you used 5 MPa as the intact yield strength of ice? Most estimates of ice yield strength are nearly an order of magnitude lower (on the order of hundreds of kPa), and the Bassis et al. (2021) paper you cited uses 0.75 MPa/1.5 MPa for uni-axial tensile strength.

L140: Delete "for their calculation methods"

L153: "representing for interactions" -> "representing interactions"

Fig 4: Could you comment more on how error may be introduced in/by the inversion? For example, if you compute basal friction coefficients at the start of the simulation, but basal slip is changing rapidly, that could change the friction coefficient which you're keeping constant in time.

L173: Put the sentence "As shown in Figure 4, we inverted…" at the end of the paragraph to make it clearer that you are commenting on observed velocities first and foremost.

Fig. 3: could you add a more descriptive y-axis label (e.g. "normalized ice thickness")

L175-180: Thanks for adding this explanation – it adds so much meaning to the manuscript!

L181: Could you be more specific about what aspects were successfully reproduced? Maybe you could say something like "our simulation successfully reproduces the decrease in ice thickness and increase in ice velocity associated with a glacier detachment."

L193: This validation section is so nice and adds so much to the paper. Thanks for including it!

L198: "Generally, the changes in englacial stress…" this sentence was already said in L188. I would also consider explaining this a little more by saying "Generally, higher ice speeds result in higher englacial stresses" or something along these lines.

Fig. 4: You should clarify in the caption that this is the inversion result for a specific timestep/snapshot before the glacier detachment. Also, why does the friction coefficient get so big at the two ends of the model domain?

L215: could be worth re-defining these two parameters very briefly (maybe inside parentheses) so that readers don't have to go all the way back. Also insert "the choice of model parameters" to show that these can be chosen arbitrarily or based on a guess.

L225: if you're going to mention monitoring for rainfall, I think you could meat up this paragraph a little more to really convince us that rainfall could be a triggering event for these glacier detachments – for example, the fact that multiple glaciers in the Sedongpu Valley detached during this event could provide even more evidence that rainfall could help trigger this positive feedback loop.

Figure 7: This is a nice figure. I would consider adding an arrow on the left side of the plot that says indicates that each row has increasing initial yield strength. That would really emphasize the visual that you only see the rapid transition to plastic flow for low initial yield strengths.

L254: "Discussions" -> "Discussion"

- The model limitations sound way nicer at this spot in the manuscript now that so many other things have been cleared up. Thanks!

Fig. 8: Consider adding an arrow indicating $t_0$ so that readers can be sure that the abrupt transition at 5 minutes is due to your changing the model physics. I would also consider adding a demarcation for Iken's bound which shows where the ratio exceeds 1 (could be a contour line, or something like that).

L269: Simplify sentence : "Kaab (2021) analyzed the force of balance of simplified, slab geometries…"

---

## Author Response (AR2)

**Reply to Reviewer 1**

"Numerical Modeling of Ice Detachment Tipping Processe: Insights from the Sedongpu Glacier, Southeastern Tibetan Plateau" by Zhang et al. (2025) uses a numerical modeling framework to demonstrate an ice weakening-basal slip feedback that results in the rapid detachment of Sedongpu Glacier which was observed in 2018.
To the authors, thank you very much for responding to all my comments in the last review.
The explanation of the science you did is so much clearer now and the manuscript is much
better organized. The sections flow much more effectively together. Now, readers can
much better appreciate the novelty and excitement of this paper! I think the paper is well
on its way to being accepted for publication, but I'd like to offer some more minor revisions.

Thank you very much for your time and efforts in improving our manuscript!

One remaining thing I would like the authors to clarify is the following. There have been some references to it in the updated manuscript, but I think it should be fleshed out a little bit more: What exactly is it that triggers the dramatic ice weakening mechanism mentioned in L213? Obviously, in real life, you cannot simply "switch on" a new piece of physics (i.e., the ice weakening-basal slip feedback) – so what natural event(s) could cause this mechanism to be activated? You have made it clear that activating this mechanism with a high initial yield strength will not trigger glacier detachment, but reasons the mechanism itself is activated are not as clear. I would love to see this a little more fleshed out in the Discussion section. It would also be nice to include a little discussion on why glacier detachment occurred at Sedongpu Valley at this particular area, yet is relatively uncommon (so far) in other glaciers of High Mountain Asia. If you don't know, it would be great to identify this as an important knowledge gap.

This is a very nice piece of advice. This is exactly why we started this project in the first place! In this paper, we know the detachment hazard could be triggered by the ice weakening-basal slip positive feedback if ice stress exceeds the yield strength. Next step, we plan to evaluate the surface and internal stress field across those glaciers in the neighboring region based on remote-sensing data and a more complicated (probably a 3D thermomechanical model) modeling approach. We could then use the tipping initial yield strength for Sedongpu to start with, and possibly adjust the tipping threshold a bit according to observations. We now add more details in the last paragraph of the Conclusion section.
"To advance this study, future efforts should extend the current two-dimensional model to three dimensions and further investigate the relationships between Iken's bound of the glacier bed and basal sliding/hydrology. By developing a more advanced numerical modeling approach, we can study glaciers in neighboring regions and estimate changes in their surface and internal ice stresses using in-situ and remote sensing observations. Subsequently, we could assess the detachment risk of surrounding glaciers, or even on a larger scale, by examining the relationships among ice stress, yield strength, surface landforms (e.g., crevasses), and basal sliding features."

Additionally, in the first set of reviews I misunderstood the authors' definition of "effective stress" because it was not defined in the text. It was not until they actually provided their definition in

the second draft that I understood where this misunderstanding came from. Many glaciologists will first associate the phrase "effective stress" or "effective pressure" to be N, the overburden pressure minus water pressure (rho g h – p_w), which is directly related to ice speeds in sliding laws such as Budd's and Schoof's, although what you are referring to as "effective stress" is technically the correct term. I think you may want to add a sentence clarifying this by emphasizing the relationship between effective stress and deviatoric stress. Ideally, I would consider referring to deviatoric stress instead of effective stress in your discussions, as this will avoid confusing readers for whom the concept of deviatoric stress is more intuitive. Alternatively, please include a small note such as "(Note: effective pressure here is defined from effective strain rate and is not related to the effective pressure N)".

The deviatoric stress has 6 independent stress component, as the Glen's law shown below.

$$\sigma'_{ij} = 2\eta\dot{\epsilon}_{ij}, \quad \eta = \frac{1}{2} A^{-1/n}(\dot{\epsilon}_e + \dot{\epsilon}_0)^{(1-n)/n},$$

The effective stress is actually a nice value representing the overall contribution of the deviatoric stress. I agree with the editor's opinion for this case, I think the current phrase/definition is clear enough.

Specific/minor comments

A small request: When adding new information (citations, clarification, new phrasings, etc.) in response to comments, please copy those changes into the response document and write what line they are at in the updated manuscript.

Thanks for this nice suggestion.

L4: "incorporating a positive feedback mechanism between ice stiffness and basal slip"

Changed.

L40: I think you can probably just cite Bassis et al. (2021)

Changed.

L42: "understandings" -> "understanding"
The introduction generally flowed so much better than it did in the first draft. I am able to understand your motivation for the science much better than before. Your explanation of glacier detachment is also much more effective than before.

Changed, and also thanks to you for your nice suggestions in the previous revision step.

L45: You can probably take out the last sentence

Removed.

L63: Delete "Glacier topography:"

Deleted.

L73: ensure that your in-text citation is formatted correctly with parentheses

Corrected.

L77: What kind of regularization does the program use? If just a simple Tikhonov, you could just say "…a linear optimization problem with a Tikhonov smoothness regularization to produce…"

Thank you for your question. We have clarified this point in the revised manuscript. The GlaTE framework applies a form of smoothness regularization consistent with the Occam's razor principle, aiming to identify the simplest solution that fits the observational and modeling constraints. This is implemented through a smoothing matrix as described by Langhammer et al. (2019), rather than explicitly using a classical Tikhonov formulation. We have revised the text to better reflect this and avoid potential confusion.

Revised text:
We estimated local ice thickness by calculating elevation differences between the pre-detachment glacier surface and the post-detachment exposed bed topography at locations where substantial ice detachment occurred. These values provided first-order estimates of ice thickness and were used as discrete constraints in the GlaTE software (Langhammer et al., 2019), which infers distributed ice thickness by optimally combining observational data with glaciological modeling in an inversion framework. The modeling component follows the method of Clarke et al. (2013), which approximates basal shear stress as a function of surface slope and apparent mass balance under a shallow-ice assumption. The inversion is formulated as a linear optimization problem with smoothness regularization, implemented via a smoothing matrix to enforce structural simplicity in the solution. We provided the estimated thickness points, a DEM, and the glacier outline as inputs to GlaTE. After obtaining the distributed ice thickness, we extracted the glacier geometry along the main centerline, which was generated following the method proposed by Kienholz et al. (2014). This flowline geometry was then used as input for the PoLIM simulations.

L80: what are the errors associated with the cross-correlation method for generating surface displacements? This could be mentioned here or in the section where you talk about other model limitations.

Thanks for your suggestions. We add below sentence in this part: "The uncertainty of surface velocity was obtained by calculating the mean displacement (5.26 m; 5.01 cm/d) from the non-glacial test areas."

L88: Thank you for providing more details about the model! These are very useful.

Thank you for your nice review.

L90: I asked in the last review about the thermal constraints on the model. Please explain in the text that you neglected thermal evolution and kept A constant, and that there is no geothermal flux (as you wrote in your responses).

We now add more details here: "Sedongpu Glacier is a typical maritime glacier in southeastern Tibet. In this study, we assume Sedongpu Glacier is temperate and set A as a constant for ice temperature close to 0 ℃ (Cuffey and Paterson, 2010) (Table 1), i.e., we do not include a temperature solver in our model. "

Figure 1d: It would be great to overlay the outline of the glacier on this panel so we can contextualize these surface elevation changes

Thanks for the advice. Fig 1d is now improved.

L102: Assuming the definition of effective strain rate in Cuffey & Paterson Eq. 3.17, your definition of effective strain rate does not make sense. If your model is 2D, it doesn't make sense that you'd have strains in 3 directions. Also, not sure where the third term (cross term) on the right-hand side comes from. Could you please clarify why you have defined effective strain rate this way?

Thanks for pointing this out. You are right, the current one is the definition for 3D. Now we correct this with the following 2D form:

$$\dot{\epsilon}_e^2 \simeq \left(\frac{\partial u}{\partial x}\right)^2 + \left(\frac{\partial v}{\partial y}\right)^2 + \frac{\partial u}{\partial x}\frac{\partial v}{\partial y} + \frac{1}{4}\left(\frac{\partial u}{\partial y}\right)^2 + \frac{1}{4}\left(\frac{\partial u}{\partial z}\right)^2 .$$

This is also mentioned in our previous paper like Zhang et al. (2013), Journal of Glaciology.

L118: Till deformation is also part of this equation – some glaciers don't slide along their beds or deform much internally, and their motion is solely due to the plastic deformation of till underneath. If you're considering till deformation to be part of basal sliding, you may want to specify this more explicitly rather than leaving it out!
How do you know that some of observed detachment is not just due to very fast till deformation? You said that glacier motion has two parts – basal sliding and internal deformation. But till deformation should be included in this. You could clarify that you are including till deformation lumped in with the friction coefficient, but you should then also address that this friction coefficient will not stay constant with time – how is this addressed?

Right. This is also a nice point. I have to agree that it is a very difficult issue. I am not sure if there is a land ice model that could properly address the problem of changing friction by coupling soft sediment. We did something similar in Greenland by using a more physical sliding law (pseudo plastic with local till model) (Zhang et al., 2024) so that we do not need to fix the basal friction

parameter during the model run, but it is still a sliding law incorporated in the ice flow model, not a separate model for deformable bed or sediment. In the revised manuscript, we change this sentence to "Factors such as soft sediments and basal meltwater lubrication will reduce the basal friction, consequently accelerating ice flow, which are not considered separately but taken as a result of changing basal frictions in the sliding law in this study." We might consider using a similar sliding law to the pseudo plastic one in the future.

Ref: Zhang, T., Colgan, W., Wansing, A., Løkkegaard, A., Leguy, G., Lipscomb, W. H., and Xiao, C.: Evaluating different geothermal heat-flow maps as basal boundary conditions during spin-up of the Greenland ice sheet, The Cryosphere, 18, 387–402, https://doi.org/10.5194/tc-18-387-2024, 2024.

L126: In the first review, I asked about Dirichlet and Neumann boundary conditions not because I didn't know what they are, but because the text at this place is misleading. Arthern et al. (2015) solves the momentum equations by using both the stress-free boundary conditions (Neumann) and observed velocities (Dirichlet). Here, you have said that Dirichlet and Neumann boundary conditions are observed and modeled ice velocities, which is a little confusing, since surface velocity is a Dirichlet condition. I would suggest reframing this closer to the way Arthern et al. (2015) defines it.

OK, we change this sentence to "where $u^D$ is the observed ice surface velocity and $u^N$ is the ice surface velocity solved in the model by applying a stress-free surface boundary condition".

Table 1: Could you please clarify why you used 5 MPa as the intact yield strength of ice? Most estimates of ice yield strength are nearly an order of magnitude lower (on the order of hundreds of kPa), and the Bassis et al. (2021) paper you cited uses 0.75 MPa/1.5 MPa for uni-axial tensile strength.

In Bassis et al. (2021) they did not state very clearly about the intact strength. In Table 2 they listed 1.5 Mpa and 3 Mpa for compression strength. In fact, the intact strength of ice varies pretty big. In Golovin et al. (2023), they said "The experimental strength of natural ice ... lies between 0.5 and 10 MPa usually". So the value of 5 MPa we use in this study is a rough estimate and we also want to use some value slightly larger than that in Bassis et al. (2021) to ensure our model not to go too crazy.
Ref: Golovin, Y.I., Samodurov, A.A., Tyurin, A.I., Rodaev, V.V., Golovin, D.Y., Vasyukov, V.M., Razlivalova, S.S. and Buznik, V.M., 2023. Ice Composites Strengthened by Organic and Inorganic Nanoparticles. Journal of Composites Science, 7(8), p.304.

L140: Delete "for their calculation methods"

Deleted.

L153: "representing for interactions" -> "representing interactions"

Changed.

Fig 4: Could you comment more on how error may be introduced in/by the inversion? For example, if you compute basal friction coefficients at the start of the simulation, but basal slip is changing rapidly, that could change the friction coefficient which you're keeping constant in time.

This is another challenging question :) Honestly, it is hard to answer. In my own opinion, the error/uncertainty in model inversion for simulating abrupt ice detachment is probably not as important as "normal" land ice projections, e.g., projecting Greenland and Antarctica Ice Sheet into the next 100 years. If you can take a look at the initMIP-Antarctica/Greenland papers, there has been clear conclusions that the initialization has large impacts on final projection results. But for this detachment case, I do not think the initial status is that important as long as the detachment mechanism is triggered. It might impact the tipping point whereby we decide if the ice stress exceeds the yield strength of ice. But it is hard to estimate, and I really do not know how to put it in the manuscript.

Ref:
Goelzer, H., Nowicki, S., Edwards, T., Beckley, M., Abe-Ouchi, A., Aschwanden, A., Calov, R., Gagliardini, O., Gillet-Chaulet, F., Golledge, N. R., Gregory, J., Greve, R., Humbert, A., Huybrechts, P., Kennedy, J. H., Larour, E., Lipscomb, W. H., Le clec'h, S., Lee, V., Morlighem, M., Pattyn, F., Payne, A. J., Rodehacke, C., Rückamp, M., Saito, F., Schlegel, N., Seroussi, H., Shepherd, A., Sun, S., van de Wal, R., and Ziemen, F. A.: Design and results of the ice sheet model initialisation experiments initMIP-Greenland: an ISMIP6 intercomparison, The Cryosphere, 12, 1433–1460, https://doi.org/10.5194/tc-12-1433-2018, 2018.
Seroussi, H., Nowicki, S., Simon, E., Abe-Ouchi, A., Albrecht, T., Brondex, J., Cornford, S., Dumas, C., Gillet-Chaulet, F., Goelzer, H., Golledge, N. R., Gregory, J. M., Greve, R., Hoffman, M. J., Humbert, A., Huybrechts, P., Kleiner, T., Larour, E., Leguy, G., Lipscomb, W. H., Lowry, D., Mengel, M., Morlighem, M., Pattyn, F., Payne, A. J., Pollard, D., Price, S. F., Quiquet, A., Reerink, T. J., Reese, R., Rodehacke, C. B., Schlegel, N.-J., Shepherd, A., Sun, S., Sutter, J., Van Breedam, J., van de Wal, R. S. W., Winkelmann, R., and Zhang, T.: initMIP-Antarctica: an ice sheet model initialization experiment of ISMIP6, The Cryosphere, 13, 1441–1471, https://doi.org/10.5194/tc-13-1441-2019, 2019.

L173: Put the sentence "As shown in Figure 4, we inverted…" at the end of the paragraph to make it clearer that you are commenting on observed velocities first and foremost.

I guess you mean "at the beginning of the paragraph"? Changed.

Fig. 3: could you add a more descriptive y-axis label (e.g. "normalized ice thickness")

I guess you mean Fig. 6? Changed as suggested.

L175-180: Thanks for adding this explanation – it adds so much meaning to the manuscript!

Thanks for your suggestion.

L181: Could you be more specific about what aspects were successfully reproduced? Maybe you could say something like "our simulation successfully reproduces the decrease in ice thickness and increase in ice velocity associated with a glacier detachment."

Great suggestion accepted.

L193: This validation section is so nice and adds so much to the paper. Thanks for including it!

Thanks for your suggestion.

L198: "Generally, the changes in englacial stress…" this sentence was already said in L188. I would also consider explaining this a little more by saying "Generally, higher ice speeds result in higher englacial stresses" or something along these lines.

Thanks for the suggestion. This sentence is changed to "Generally, higher ice flow velocities result in greater englacial stresses, increasing the vulnerability of ice regions to detachment instability."

Fig. 4: You should clarify in the caption that this is the inversion result for a specific timestep/snapshot before the glacier detachment. Also, why does the friction coefficient get so big at the two ends of the model domain?

A sentence is added in the caption: The inversion is based on velocities observed by remote sensing from 2015 to 2018. Regarding the marginal friction, it is actually the same question asked by another reviewer in the last review. So I just copy over the reply here:

We do prescribe an initial $\beta$ = 1e3 Pa m$^{-1}$ yr. At the glacier head and terminus, we use a Dirichlet boundary condition in the velocity solver, so $\beta$ will not be updated during the initialization. Here I present three sensitivity plots for initial $\beta$ = 1e2, 1e3 and 1e4 Pa m$^{-1}$ yr. We can see clearly that the initial value has some impacts on grids close to the head and terminus, but the majority of glacier is not affected. To avoid confusion, we do not plot $\beta$ for the head and terminus grids in the revised manuscript.

[Figure]

Figure 1: initial $\beta$ = 100 Pa m$^{-1}$ yr

Figure 2: initial $\beta$ = 1000 Pa m$^{-1}$ yr

Figure 3: initial $\beta$ = 10000 Pa m$^{-1}$ yr

L215: could be worth re-defining these two parameters very briefly (maybe inside parentheses) so that readers don't have to go all the way back. Also insert "the choice of model parameters" to show that these can be chosen arbitrarily or based on a guess.

"prescribed minimum yield stress" and "prescribed minimum viscosity" are added. "the choice of

model parameters" is inserted.

L225: if you're going to mention monitoring for rainfall, I think you could meat up this paragraph a little more to really convince us that rainfall could be a triggering event for these glacier detachments – for example, the fact that multiple glaciers in the Sedongpu Valley detached during this event could provide even more evidence that rainfall could help trigger this positive feedback loop.

Right. Despite some record shows that there was heavy rainfall before the Sedongpu detachment, we still lack strong evidence in between. To avoid confusion, I remove this part in the revised manuscript.

Figure 7: This is a nice figure. I would consider adding an arrow on the left side of the plot that says indicates that each row has increasing initial yield strength. That would really emphasize the visual that you only see the rapid transition to plastic flow for low initial yield strengths.

Thanks for the suggestion, but I think I would just skip this arrow idea. The current text in three rows have shown clearly the changes of initial yield strength. But I add an additional sentence in the caption to emphasize this: "The rapid transition to plastic flow occurs for low initial yield strengths."

L254: "Discussions" -> "Discussion"
- The model limitations sound way nicer at this spot in the manuscript now that so many other things have been cleared up. Thanks!

Changed, and thank you for your nice suggestions.

Fig. 8: Consider adding an arrow indicating $t_0$ so that readers can be sure that the abrupt transition at 5 minutes is due to your changing the model physics. I would also consider adding a demarcation for Iken's bound which shows where the ratio exceeds 1 (could be a contour line, or something like that).

Fig. 8 is improved as suggested. The white circles mark the initial occurrence of the Iken ratio along the x-axis after detachment begins

[Figure]

L269: Simplify sentence : "Kaab (2021) analyzed the force of balance of simplified, slab geometries…"

Changed.

**Reply to Reviewer 2**

Overview:

The authors have responded well to comments from both reviewers and correspondingly revised this manuscript. I find the paper to be much clearer and polished, and it will be a valuable contribution to the glaciological literature to further understanding of rapid glacier detachment processes. Below you will find specific comments listing a few corrections and clarifications, but overall I feel this work is in good shape and should be published following these technical corrections.

Thank you very much for your time and efforts in improving our manuscript!

Specific comments:

Line 29: criterion (singular, not plural criteria)

Changed.

Figure 1: This figure is great, nice improvements. One comment - in panel d, I think this plot shows the difference between 2018-2015 (with surface elevation lowering over the glacier and increasing downstream). This is labeled incorrectly as 2015-2018.

Thanks for the advice. Fig 1d is now improved.

Figure 2: Change punctuation to be consistent following panel labels, also " ... black curve represents "

Changed. Thanks.

Line 153: " representing interactions " (remove the word " for " )

Removed.

Line 185 and elsewhere: kPa is the conventional abbreviation for kilopascals (not capitalized KPa)

Corrected.

Line 191: What happens to the oscillations with a smaller (or zero) minimum ice thickness imposed instead of 1m?

We tried and the model will easily go collapse.

Line 196: suggested rephrasing: " ..., which provides strong evidence for validation of our model

results.”

Changed.

Paragraph beginning line 197: This appears to be a repeated but slightly different version of the preceding paragraph. Correct the manuscript to only include only one or the other.

Thanks for point this out. Corrected.

Line 240: Remove word “primarily” – this suggests one main driver, but here the point is that there are multiple factors.

Removed.

Line 266: Add a citation or multiple citations for sliding laws that depend on hydrology, rather than just “Schoof sliding law” – or else leave out any mention of specific sliding laws

Changed. We now add a "Hoffman and Price (2014)" citation, and remove the "Schoof sliding law".

Line 269: Suggested rephrasing: “Kaab et al. (2021) analyzed…”

Changed.

Line 288: Instead of writing what you may do, I suggest using language to recommend: “To advance this research, future efforts should extend…”

Changed.

Section 6, Conclusions: You may want to consider including a bit more detailed summary of your specific findings here, as an easy reference for somebody skimming the paper to succinctly find a summary of what you did and what results you found.

We now add an additional sentence in Conclusion to give some of the details of our model findings: From the model results, we find that glacier ice detachment occurs when the initial yield strength drops to approximately 430 kPa, indicating that the ice's mechanical properties are critical in triggering abrupt collapse when mechanical stress exceeds critical failure thresholds.